# Refined Learning Bounds for Kernel and Approximate $k$-Means

**Yong Liu**[1,2]
[1]Gaoling School of Artificial Intelligence, Renmin University of China, Beijing, China
[2]Beijing Key Laboratory of Big Data Management and Analysis Methods, Beijing, China
liuyonggsai@ruc.edu.cn

## Abstract

Kernel $k$-means is one of the most popular approaches to clustering and its theoretical properties have been investigated for decades. However, the existing state-of-the-art risk bounds are of order $\mathcal{O}(k/\sqrt{n})$, which do not match with the stated lower bound $\Omega(\sqrt{k/n})$ in terms of $k$, where $k$ is the number of clusters and $n$ is the size of the training set. In this paper, we study the statistical properties of kernel $k$-means and Nyström-based kernel $k$-means, and obtain optimal clustering risk bounds, which improve the existing risk bounds. Particularly, based on a refined upper bound of Rademacher complexity [21], we first derive an optimal risk bound of rate $\mathcal{O}(\sqrt{k/n})$ for empirical risk minimizer (ERM), and further extend it to general cases beyond ERM. Then, we analyze the statistical effect of computational approximations of Nyström kernel $k$-means, and prove that it achieves the same statistical accuracy as the original kernel $k$-means considering only $\Omega(\sqrt{nk})$ Nyström landmark points. We further relax the restriction of landmark points from $\Omega(\sqrt{nk})$ to $\Omega(\sqrt{n})$ under a mild condition. Finally, we validate the theoretical findings via numerical experiments.

## 1 Introduction

Clustering, a fundamental data mining task, is used in numerous applications including web search, medical imaging, gene expression analysis, social network analysis and recommendation systems [56, 55, 23, 42]. $k$-means is arguably one of the most popular approaches to clustering, producing clusters with piece-wise linear boundaries. Its kernel version, which employs a nonlinear distance function, has the ability to find clusters of varying densities and distributions, greatly improving the flexibility of the approach [18, 53, 38, 37, 39, 60, 29, 36, 35, 61].

To understand (kernel) $k$-means and guide the development of new clustering algorithms, researchers have investigated its theoretical properties for decades. The consistency of the empirical minimizer was demonstrated by [45, 47, 1]. Rates of convergence and non-asymptotic performance bounds were considered by [46, 13, 33, 7, 32, 14, 20]. Most of the proposed risk bounds are dependent upon the dimension of the hypothesis space. For example, Bartlett et al. [7] provided, under certain mild assumptions, a clustering risk bound of order $\mathcal{O}(\sqrt{kd/n})$, where $d$ is the dimension of the hypothesis space and $n$ is the size of the training set. However, the hypothesis space of kernel $k$-means is typically an infinite-dimensional Hilbert space, such as the reproducing kernel Hilbert space (RKHS) associated with Gaussian kernels [50]. Thus, the existing theoretical analysis of $k$-means are not usually suitable for explaining its kernel version. Recently, [20, 16, 10, 41, 3, 27, 24, 9] extended the previous results, and provided dimension-independent bounds for kernel $k$-means. As shown in [16], if the feature map associated with the kernel function satisfies $\|\Phi\| \leq 1$, then the clustering risk bounds are of order $\mathcal{O}(k/\sqrt{n})$. These clustering risk bounds for kernel $k$-means are usually linearly

dependent on the number of clusters $k$. However, the number of clusters $k$ may be very large in some domains, such as social networks and recommendation systems. Thus, from a theoretical perspective, these existing bounds of $\mathcal{O}(k/\sqrt{n})$ do not match with the stated lower bound $\Omega(\sqrt{k/n})$ in $k$ [7].

Although kernel $k$-means is one of the most popular clustering methods, it requires the computation of a $n \times n$ kernel matrix. As for other kernel methods, this becomes unfeasible for large-scale problems, and thus deriving approximate computations, such as partial decompositions [6, 28], random projection [16, 15], Nyström approximations [19, 11, 9, 43, 53, 58, 57], and random feature approximations [48, 12, 5, 49, 34, 31], has become the subject of numerous recent works. However, few of these optimization-based methods focused on the underlying excess risk problem. To the best of our knowledge, the only two results providing excess risk guarantees for approximate kernel $k$-means are [16] and [9]. In [16], Devroye and Lugosi considered the excess clustering risk when the approximate Hilbert space is obtained using Gaussian projections. In [9], Calandriello and Rosasco showed that, when sampling $\Omega(\sqrt{n})$ Nyström landmarks, the excess risk bound can reach $\mathcal{O}(k/\sqrt{n})$. The excess risk bounds of [9] and [16] are both linearly dependent on $k$ and thus do not match with the theoretical lower bound [7].

In the recent work [21], the authors showed that the Rademacher complexity of the $k$-valued function class of Lipschitz continuity with respect to the $L_\infty$ norm can be bounded by the maximum Rademacher complexity of the restriction of the function class along each coordinate, times a factor of $\mathcal{O}(\sqrt{k})$. Although it may be not very difficult to use the result of [21] for kernel $k$-means, the optimal bound of $\mathcal{O}(\sqrt{k/n})$ for kernel $k$-means has never been given before. Moreover, we creatively extend the results of kernel $k$-means to the approximate one. Our major contributions include two parts:

1) A (nearly) optimal excess clustering risk bound of rate $\tilde{\mathcal{O}}(\sqrt{k/n})$[1] is proposed for empirical risk minimization (ERM) (see Theorem 1). To the best of our knowledge, this is the first (nearly) optimal excess risk bound for kernel $k$-means in terms of both $k$ and $n$. Beyond ERM, we further extend the result of Theorem 1 to general cases (see Theorem 2 and Theorem 3).

2) A (nearly) optimal excess risk bound for Nyström kernel $k$-means is also obtained when sampling $\Omega(\sqrt{nk})$ points (see Theorem 4). We further relax the restriction of landmark points from $\Omega(\sqrt{nk})$ to $\Omega(\sqrt{n})$ (see Theorem 5) and extend it to general cases (see Theorem 6 and Theorem 7). This result shows that we can use the Nyström method to improve the effectiveness of kernel $k$-means, while guaranteeing the optimal generalization performance.

The rest of the paper is organized as follows. In Section 2, we introduce some notations and provide an overview of kernel $k$-means. In Section 3, we provide nearly optimal excess risk bounds. In Section 4, we quantify the statistical effect of computational approximations of the Nyström-based kernel $k$-means. In Section 5, we validate our theoretical findings by performing experiments on both simulated and real data. We end in Section 6 with conclusion. All the detailed proofs are deferred to the Appendix.

## 2 Background

In this section, we will introduce some notations and provide a brief introduction of kernel $k$-means. Please refer to [18, 9] for more details.

### 2.1 Notations

Assume $\mathbb{P}$ is a (unknown) distribution on $\mathcal{X}$, and $\mathcal{S} = \{\mathbf{x}_i\}_{i=1}^n \in \mathcal{X}$ is a set of $n$ samples drawn i.i.d. from $\mathbb{P}$. The *empirical* distribution $\mathbb{P}_n$ is denoted as $\mathbb{P}_n(\boldsymbol{x}) = \frac{1}{n}$ if $\mathbf{x} \in \mathcal{S}$, otherwise 0. Let $\kappa : \mathcal{X} \times \mathcal{X} \to \mathbb{R}$ be a mercer kernel [51], and $\mathcal{H}$ be its associated RKHS [52], which is the completion of the linear span of the set of functions:

$$\mathcal{H} = \overline{\text{span}\{\kappa(\mathbf{x}, \cdot), \mathbf{x} \in \mathcal{X}\}}.$$

We denote the Cartesian product of $\mathcal{H}$ by $\mathcal{H}^k = \otimes_{i=1}^k \mathcal{H}$. We use the *feature map* $\psi : \mathcal{X} \to \mathcal{H}$ to map $\mathcal{X}$ into the Hilbert space $\mathcal{H}$, and assume that $\mathcal{H}$ is separable, such that for any $\mathbf{x} \in \mathcal{X}$, we have

---

[1]$\tilde{\mathcal{O}}$ hides logarithmic terms.

$\Phi_{\mathbf{x}} = \psi(\mathbf{x})$. Intuitively, in the rest of the paper, the reader can assume that $\Phi_{\mathbf{x}} \in \mathbb{R}^d$ with $d \gg n$ or even infinite. From here on, we will denote the inner product of $\mathcal{H}$ by $\langle \cdot, \cdot \rangle$, and the associated norm by $\| \cdot \|$, and assume that $\|\Phi_{\mathbf{x}}\| \leq 1$ for any $\mathbf{x} \in \mathcal{X}$. We let

$$\mathcal{D} = \{\Phi_i = \psi(\mathbf{x}_i)\}_{i=1}^n,$$

and denote $[\mathbf{K}]_{i,j} = \kappa(\mathbf{x}_i, \mathbf{x}_j) = \langle \Phi_i, \Phi_j \rangle$ as the kernel matrix.

The notations $\mu = \mathcal{O}(\nu)$ and $\mu = \Omega(\nu)$ mean that there exist constants $c, c_1, c_2$ such that $\mu \leq c\nu$ and $c_1\nu \leq \mu \leq c_2\nu$, respectively. We use $\tilde{\mathcal{O}}$ and $\tilde{\Omega}$ to hide logarithmic terms.

## 2.2 Kernel $k$-Means

In this paper, we aim at partitioning the given dataset into $k$ disjoint *clusters*, each characterized by its *centroid* $\mathbf{c}_j$. The Voronoi cell associated with a centroid $\mathbf{c}_j$ is defined as [9]

$$\mathcal{C}_j := \left\{ i : j = \arg\min_{s=1,\dots,k} \|\Phi_i - \mathbf{c}_s\|^2 \right\}.$$

Let $\mathbf{C} = [\mathbf{c}_1, \dots, \mathbf{c}_k]$ be a collection of $k$ centroids from $\mathcal{H}^k$. In this paper, we focus on the so-called *kernel $k$-means* clustering, by minimizing the *empirical squared norm criterion*

$$\mathcal{W}(\mathbf{C}, \mathbb{P}_n) := \frac{1}{n} \sum_{i=1}^n \min_{j=1,\dots,k} \|\Phi_i - \mathbf{c}_j\|^2 \tag{1}$$

over all possible choices of cluster centers $\mathbf{C} \in \mathcal{H}^k$. From [18, 9], we know that $\mathcal{W}(\mathbf{C}, \mathbb{P}_n)$ can be written as

$$\mathcal{W}(\mathbf{C}, \mathbb{P}_n) := \frac{1}{n} \min_{\mathbf{C} \in \mathcal{H}^k} \sum_{j=1}^k \sum_{i \in \mathcal{C}_j} \left\| \Phi_i - \frac{1}{|\mathcal{C}_j|} \sum_{t \in \mathcal{C}_j} \Phi_t \right\|^2$$

$$= \frac{1}{n} \min_{\mathbf{C} \in \mathcal{H}^k} \sum_{j=1}^k \sum_{i \in \mathcal{C}_j} \left( \kappa(\mathbf{x}_i, \mathbf{x}_i) - \frac{2}{|\mathcal{C}_j|} \sum_{t \in \mathcal{C}_j} \kappa(\mathbf{x}_i, \mathbf{x}_t) + \frac{1}{|\mathcal{C}_j|^2} \sum_{t, t' \in \mathcal{C}_j} \kappa(\mathbf{x}_t, \mathbf{x}_{t'}) \right).$$

The *empirical risk minimizer* (*ERM*) is defined as

$$\mathbf{C}_n := \arg\min_{\mathbf{C} \in \mathcal{H}^k} \mathcal{W}(\mathbf{C}, \mathbb{P}_n). \tag{2}$$

The performance of a clustering scheme given by the collection $\mathbf{C} = [\mathbf{c}_1, \dots, \mathbf{c}_k] \in \mathcal{H}^k$ of cluster centers is usually measured by the *expected squared norm criterion* or *expected clustering risk*

$$\mathcal{W}(\mathbf{C}, \mathbb{P}) := \int \min_{j=1,\dots,k} \|\Phi_{\mathbf{x}} - \mathbf{c}_j\|^2 d\mathbb{P}(\mathbf{x}).$$

Given a $\mathbf{C} \in \mathcal{H}^k$, let $f_{\mathbf{C}} = (f_{\mathbf{c}_1}, \dots, f_{\mathbf{c}_k})$ be a *$k$-valued* function of the collection $\mathbf{C}$ with $f_{\mathbf{c}_j}(\mathbf{x}) = \|\Phi_{\mathbf{x}} - \mathbf{c}_j\|^2$. Let $\varphi : \mathbb{R}^k \to \mathbb{R}$ be a minimum function. From the definition of $\varphi(f_{\mathbf{C}}(\mathbf{x})) = \min(f_{\mathbf{c}_1}(\mathbf{x}), \dots, f_{\mathbf{c}_k}(\mathbf{x}))$, one can see that the empirical and expected squared norm criteria can be respectively written as

$$\mathcal{W}(\mathbf{C}, \mathbb{P}_n) := \frac{1}{n} \sum_{i=1}^n \varphi(f_{\mathbf{C}}(\mathbf{x}_i)) \text{ and } \mathcal{W}(\mathbf{C}, \mathbb{P}) := \int \varphi(f_{\mathbf{C}}(\mathbf{x})) d\mathbb{P}(\mathbf{x}).$$

In this paper, we consider bounding the *excess clustering risk* $\mathcal{E}(\mathbf{C}_n)$ of the empirical risk minimizer [16]:

$$\mathcal{E}(\mathbf{C}_n) := \mathbb{E}_{\mathcal{D}}[\mathcal{W}(\mathbf{C}_n, \mathbb{P})] - \mathcal{W}^*(\mathbb{P}),$$

where $\mathcal{W}^*(\mathbb{P}) = \inf_{\mathbf{C} \in \mathcal{H}^k} \mathcal{W}(\mathbf{C}, \mathbb{P})$ is the *optimal* clustering risk. In the following, we will ignore the subscript $\mathcal{D}$ if the input dataset $\mathcal{D}$ is clear.

## 2.3 The Existing Excess Clustering Risk Bounds

According to [7], we know that there exists a collection of centroids $\mathbf{C}_{lb} \in \mathcal{H}^k$, a constant $c$, and a distribution $\mathbb{P}$ with $\|\Phi_{\mathbf{x}}\| \leq 1$ for any $\mathbf{x} \in \mathcal{X}$, such that

$$\mathbb{E}[\mathcal{W}(\mathbf{C}_{lb}, \mathbb{P})] - \mathcal{W}^*(\mathbb{P}) \geq c\sqrt{\frac{k^{1-4/d}}{n}}.$$

Note that $d$ is the dimension of $\Phi_{\mathbf{x}}$, which is usually very large or even infinite. Thus, the lower bound of kernel $k$-means is $\Omega(\sqrt{k/n})$. However, most of the existing risk bounds proposed for kernel $k$-means are $\mathcal{O}(k/\sqrt{n})$ [16, 10, 41, 20, 9], for example:

**Lemma 1** ([16], Theorem 2.1). *If $\|\Phi_{\mathbf{x}}\| \leq 1$ for any $\mathbf{x} \in \mathcal{X}$, then there exists a constant $c$ such that*

$$\mathbb{E}[\mathcal{W}(\mathbf{C}_n, \mathbb{P})] - \mathcal{W}^*(\mathbb{P}) \leq c\frac{k}{\sqrt{n}},$$

*where $\mathbf{C}_n$ is the ERM of $\mathcal{W}(\mathbf{C}, \mathbb{P}_n)$ defined in* (2).

Note that the number of clusters $k$ may be very large for fine-grained analyses in social networks or recommendation systems. This leaves us with the question: is it possible to prove a bound of rate $\sqrt{k/n}$, which is (nearly) optimal in terms of both $k$ and $n$? In this paper, we attempt to answer this.

## 3 Main Results

In this section, we will provide nearly optimal excess risk bounds for kernel $k$-means. There are very few works focus on the underlying excess risk problem for kernel $k$-means. To the best of our knowledge, there are only two results [16, 9] providing excess risk bounds for kernel $k$-means or approximate kernel $k$-means. However, these bounds of [16, 9] are all linearly dependent on $k$. Based on a recently improvement of the upper bound of Rademacher complexity of $L$-Lipschitz with respect to the $L_\infty$ norm [21], we derive a (nearly) optimal excess risk bound of linearly dependent on $\sqrt{k}$.

**Theorem 1.** *If $\forall \mathbf{x} \in \mathcal{X}, \|\Phi_{\mathbf{x}}\| \leq 1$, then for any $\delta \in (0,1)$, there exists a constant $c$, and with probability at least $1 - \delta$, we have,*

$$\mathbb{E}[\mathcal{W}(\mathbf{C}_n, \mathbb{P})] - \mathcal{W}^*(\mathbb{P}) \leq c\left(\sqrt{\frac{k}{n}} \log^2\left(\sqrt{n}\right) + \sqrt{\frac{\log\frac{1}{\delta}}{n}}\right).$$

From Theorem 1, we know that

$$\mathbb{E}[\mathcal{W}(\mathbf{C}_n, \mathbb{P})] - \mathcal{W}^*(\mathbb{P}) \leq \tilde{\mathcal{O}}\left(\sqrt{\frac{k}{n}}\right),$$

which matches the theoretical lower bound $\Omega(\sqrt{k/n})$ when $d$ is large [7]. Thus, our proposed bound is **(nearly) optimal**.

**Remark (Fast Rates).** Some results suggest that the learning rate of kernel $k$-means can reach $\mathcal{O}(k/n)$ under certain assumptions on the distribution. Chou [13] pointed out that, if continuous densities of distribution satisfy certain regularity properties, the expected excess risk is of rate $\mathcal{O}(k/n)$. An improved result was obtained by [3], who proved that the learning rate can reach $\mathcal{O}(k/n)$ for any distribution supported on a finite set. Levrard [27] further showed that, if the distribution satisfies a margin condition, the learning rate can also reach $\mathcal{O}(k/n)$. Based on the notion of local Rademacher complexity, the expected excess risk has a rate faster than $\mathcal{O}(k/\sqrt{n})$ given in [24, 30]. However, as pointed out, these conditions are difficult to verify in general. Moreover, these expected excess risk bounds are linearly dependent on $k$. In the future, we will consider studying whether it is possible to prove a bound of $\mathcal{O}(\sqrt{k}/n)$ under certain strict assumptions.

### 3.1 Further Results: Beyond ERM

So far we have provided guarantees for $\mathbf{C}_n$, that is, the optimal ERM in $\mathcal{H}^k$. Note that obtaining the optimal ERM $\mathbf{C}_n$ is a NP-hard problem in general [2]. In the following, we will consider the risk bound for a general $\tilde{\mathbf{C}}_n$, which only requires that its empirical squared norm criterion is not far from that of $\mathbf{C}_n$.

**Theorem 2.** *If* $\forall \mathbf{x} \in \mathcal{X}, \|\Phi_{\mathbf{x}}\| \leq 1$ *and*

$$\mathbb{E}\big[\mathcal{W}(\tilde{\mathbf{C}}_n, \mathbb{P}_n) - \mathcal{W}(\mathbf{C}_n, \mathbb{P}_n)\big] \leq \zeta,$$

*then for any* $\delta \in (0, 1)$*, there exists a constant* $c$ *and, with probability at least* $1 - \delta$*, we have*

$$\mathbb{E}[\mathcal{W}(\tilde{\mathbf{C}}_n, \mathbb{P})] - \mathcal{W}^*(\mathbb{P}) \leq c\sqrt{\frac{k}{n}} \log^2\left(\sqrt{n}\right) + c\sqrt{\frac{\log \frac{1}{\delta}}{n}} + \zeta.$$

From the above theorem, one can see that if the discrepancy between the empirical squared norm criterion of $\tilde{\mathbf{C}}_n$ and $\mathbf{C}_n$ is small, that is $\zeta \leq \mathcal{O}(\sqrt{k/n})$, the risk bound of $\tilde{\mathbf{C}}_n$ is (nearly) optimal.

### 3.2 Further Results: $k$-means++

Lloyd's algorithm [40] is the most popular $k$-means algorithm and when coupled with a careful $k$-means++ seeding [4], a good approximate solution $\tilde{\mathbf{C}}_n$ can be obtained. Recently, based on a simple combination of $k$-means++ sampling and a local search strategy, an improved $k$-means++ algorithm was proposed [25]. It was shown that the empirical squared norm criterion of $\tilde{\mathbf{C}}_n$ can be up to a constant factor from the optimal empirical solution. For the completeness, we briefly describe the improved $k$-means++ in the following, please refer to [25] for more details.

1: If $|\mathcal{C}| < k$, add a sampled point $\mathbf{x} \in \mathcal{S}$ with probability

$$\frac{\text{cost}(\{\psi(\mathbf{x})\}, \mathcal{C})}{\sum_{\mathbf{x} \in \mathcal{S}} \text{cost}(\{\psi(\mathbf{x})\}, \mathcal{C})}, \text{ where } \text{cost}(\mathcal{P}, \mathcal{C}) = \sum_{\mathbf{x}_i \in \mathcal{P}} \min_{\mathbf{c} \in \mathcal{C}} \|\Phi_i - \mathbf{c}\|,$$

and add $\psi(\mathbf{x})$ to $\mathcal{C}$.

2: If $|\mathcal{C}| \geq k$, sample $\mathbf{x} \in \mathcal{S}$ with probability $\frac{\text{cost}(\{\psi(\mathbf{x})\}, \mathcal{C})}{\sum_{\mathbf{x} \in \mathcal{S}} \text{cost}(\{\psi(\mathbf{x})\}, \mathcal{C})}$, check whether there exists a point $\mathbf{c} \in \mathcal{C}$ such that

$$\text{cost}(\mathcal{S}, \mathcal{C} \backslash \{\mathbf{c}\} \cup \{\psi(\mathbf{x})\}) < \text{cost}(\mathcal{S}, \mathcal{C}).$$

If this is the case, we replace $\mathbf{c}$ by the point in $\mathcal{C}$ that reduces the cost function by the largest amount.

Note that we use the algorithm from [25] for kernel $k$-means by replacing the Euclidean distance $\|\mathbf{x}_i - \mathbf{x}_j\|^2$ with $\|\Phi_i - \Phi_j\|_{\mathcal{H}}^2 = \kappa(\mathbf{x}_i, \mathbf{x}_i) - 2\kappa(\mathbf{x}_i, \mathbf{x}_j) + \kappa(\mathbf{x}_j, \mathbf{x}_j)$.

**Lemma 2** ([25]). *If* $\mathbf{C}_n^{\mathcal{A}}$ *is returned by the improved $k$-means++ algorithm with a local search strategy [25], then*

$$\mathbb{E}_{\mathcal{A}}[\mathcal{W}(\mathbf{C}_n^{\mathcal{A}}, \mathbb{P}_n)] \leq \beta \cdot \mathcal{W}(\mathbf{C}_n, \mathbb{P}_n),$$

*where* $\beta$ *is a constant and* $\mathcal{A}$ *is the randomness derived from the $k$-means++ initialization.*

In the following, we derive a risk bound for $\mathbf{C}_n^{\mathcal{A}}$.

**Theorem 3.** *If* $\forall \mathbf{x} \in \mathcal{X}, \|\Phi_{\mathbf{x}}\| \leq 1$*, and* $\mathbf{C}_n^{\mathcal{A}}$ *is returned by the improved $k$-means++ algorithm with a local search strategy [25], then for any* $\delta \in (0, 1)$*, with a probability at least* $1 - \delta$*, we have*

$$\mathbb{E}_{\mathcal{D}}\left[\mathbb{E}_{\mathcal{A}}[\mathcal{W}(\mathbf{C}_n^{\mathcal{A}}, \mathbb{P})]\right] \leq \tilde{\mathcal{O}}\left(\sqrt{\frac{k}{n}} + \mathcal{W}^*(\mathbb{P})\right).$$

The above result implies that if the optimal clustering risk $\mathcal{W}^*(\mathbb{P})$ is small, the risk of $\mathcal{W}(\mathbf{C}_n^{\mathcal{A}}, \mathbb{P})$ can reach $\tilde{\mathcal{O}}(\sqrt{k/n})$.

## 4 Risk Analysis of Nyström Kernel $k$-Means

Kernel $k$-means is one of the most popular clustering methods. However, it requires the computation of a $n \times n$ kernel matrix. This renders it non-scalable to large datasets that contain more than a few

tens of thousands of points. In particular, simply constructing and storing the kernel matrix $\mathbf{K}$ takes $O(n^2)$ time and space.

The Nyström method [19] is a popular method for approximating the kernel matrix. The properties of Nyström approximations for kernel $k$-means have recently been studied in [11, 15, 43, 9, 53, 59]. However, most of these works focus on the computation area. To the best of our knowledge, the only study providing excess risk guarantees for the Nyström kernel $k$-means is [9]. However, its excess risk bound is linearly dependent on $k$. In the following, we will improve it from $k$ to $\sqrt{k}$.

### 4.1 Nyström Kernel $k$-Means

To derive the excess risk bound of Nyström kernel $k$-means, we first briefly introduce some notations. Given a dataset $\mathcal{D} = \{\Phi_i\}_{i=1}^n$, we use

$$\mathcal{I} = \{\Phi_i\}_{i=1}^m \subseteq \mathcal{D}$$

as a collection of landmark points to replace $\mathcal{D}$. Let $\mathcal{H}_m$ be a linear span of $\mathcal{I} = \{\Phi_i\}_{i=1}^m$,

$$\mathcal{H}_m = \text{span}\left\{\sum_{i=1}^m \alpha_i \Phi_i, \alpha_i \in \mathbb{R}, \Phi_i \in \mathcal{I}\right\},$$

and $\mathcal{H}_m^k = \otimes_{i=1}^k \mathcal{H}_m$ be its Cartesian product. The Nyström kernel $k$-means, i.e., the approximate kernel $k$-means over $\mathcal{H}_m^k$, can be written as [9]:

$$\mathbf{C}_{n,m} = \arg\min_{\mathbf{C} \in \mathcal{H}_m^k} \frac{1}{n} \sum_{i=1}^n \min_{j=1,\dots,k} \|\Phi_i - \mathbf{c}_j\|^2. \tag{3}$$

The centroids $\mathbf{C}_{n,m}$ are still point in $\mathcal{H}_m \subset \mathcal{H}$. Let $\mathbf{K}_{m,m} \in \mathbb{R}^{m \times m}$ be the empirical kernel matrix between all points in $\mathcal{I}$, and its eigen-decomposition is $\mathbf{K}_{m,m} = \mathbf{U}\boldsymbol{\Lambda}\mathbf{U}$. From [9], we can search over $\tilde{\mathbf{C}}_{n,m} \in \mathbb{R}^{m \times k}$ instead of searching over $\mathbf{C} \in \mathcal{H}_m^k$, that is,

$$\tilde{\mathbf{C}}_{n,m} = \arg\min_{\tilde{\mathbf{C}} \in \mathbb{R}^{m \times k}} \frac{1}{n} \sum_{i=1}^n \min_{j=1,\dots,n} \|\tilde{\Phi}_i - \tilde{\mathbf{c}}_j\|^2, \tag{4}$$

where $\tilde{\Phi}_i := \boldsymbol{\Lambda}^{-1/2}\mathbf{U}^\mathrm{T}\boldsymbol{\Phi}_m^\mathrm{T}\Phi_i$, $\tilde{\mathbf{c}}_j := \boldsymbol{\Lambda}^{-1/2}\mathbf{U}^\mathrm{T}\boldsymbol{\Phi}_m^\mathrm{T}\mathbf{c}_j$, $\boldsymbol{\Phi}_m = [\Phi_{\pi(1)}, \dots, \Phi_{\pi(m)}]$, $\pi(i) \in [1, m]$. We can use any $k$-means algorithms to solve Eq.4, and then use the reverse of the relationship $\tilde{\mathbf{c}}_j := \boldsymbol{\Lambda}^{-1/2}\mathbf{U}^\mathrm{T}\boldsymbol{\Phi}_m^\mathrm{T}\mathbf{c}_j$ to bring back the solution to $\mathcal{H}_m$, i.e., $\mathbf{C}_{n,m} = \boldsymbol{\Phi}_m\mathbf{U}\boldsymbol{\Lambda}^{-1/2}\tilde{\mathbf{C}}_{n,m}$. This can be done in $\mathcal{O}(nm)$ space and $\mathcal{O}(nmkt + nm^2)$ time using $t$ steps of Lloyd's algorithm for $k$ clusters [40]. Please refer to [9] for more details.

### 4.2 Excess Risk Bound of Nyström Kernel $k$-Means

Denote with $\Xi = \text{Tr}(\mathbf{K}^\mathrm{T}(\mathbf{K} + \mathbf{I})^{-1})$ the so-called effective dimension of $\mathbf{K}$ [49, 9]. Note that

$$\text{Tr}\left(\mathbf{K}^\mathrm{T}(\mathbf{K} + \mathbf{I})^{-1}\right) \leq \text{Tr}\left(\mathbf{K}^\mathrm{T}(\mathbf{K})^+\right),$$

so we can obtain that $\Xi \leq \text{Rank}(\mathbf{K})$. Thus, the effective dimension $\Xi$ can be seen as a soft version of the rank.

**Theorem 4.** *If $\forall \mathbf{x} \in \mathcal{X}, \|\Phi_\mathbf{x}\| \leq 1$, and the size of a uniform sampling is*

$$m \geq \Omega\left(\frac{\sqrt{n}\log(1/\delta)\min(k, \Xi)}{\sqrt{k}}\right),$$

*then, with probability at least $1 - \delta$, we have*

$$\mathbb{E}[\mathcal{W}(\mathbf{C}_{n,m}, \mathbb{P})] - \mathcal{W}^*(\mathbb{P}) \leq \mathcal{O}\left(\sqrt{\frac{k}{n}}\log\left(\frac{n}{\delta}\right)\right).$$

Note that
$$\frac{\sqrt{n}\min(k,\Xi)}{\sqrt{k}} \le \sqrt{nk}.$$

Thus, from a statistical point, Theorem 4 shows that when sampling $\tilde{\Omega}(\sqrt{nk})$ points, the Nyström kernel $k$-means achieves the same excess risk as the exact one does. This result demonstrates that we can improve the computational aspect of kernel $k$-means using Nyström embedding, while maintaining **optimal** generalization guarantees.

**Remark.** Calandriello and Rosasco [9] have reported that if $m \ge \tilde{\Omega}(\sqrt{n})$, an excess risk bound of rate $\tilde{\mathcal{O}}(k/\sqrt{n})$ for Nyström kernel $k$-means can be obtained, which seems to be better than our $\tilde{\Omega}(\sqrt{nk})$. However, it should be noted that the risk bound in [9] is linearly dependent on $k$, while ours is linearly dependent on $\sqrt{k}$. From the proof of Lemma 10, if we want to obtain a risk of linear dependence on $k$, we only need
$$m \ge \Omega\left(\frac{\sqrt{n}\log(1/\delta)\min(k,\Xi)}{k}\right) = \tilde{\Omega}(\sqrt{n}),$$

which is the same as [9]. In the following, we will show that we can relax the restriction of landmark points under a mild condition.

## 4.3 Further Results: Reducing the Sampling Points

From Theorem 4, we know that we need $\tilde{\Omega}(\sqrt{nk})$ sampling points to guarantee the nearly optimal rate for approximating kernel $k$-means. In the following, we show how to reduce the sampling points from $\tilde{\Omega}(\sqrt{nk})$ to $\tilde{\Omega}(\sqrt{n})$ under a basic assumption on the eigenvalues of the kernel matrix.

**Theorem 5.** *Let $\lambda_i$ be the $i$-th eigenvalue of the kernel matrix $\mathbf{K}$, $i = 1, \dots, n$, and $\lambda_{i+1} \le \lambda_i$. If $\forall \mathbf{x} \in \mathcal{X}, \|\Phi_{\mathbf{x}}\| \le 1$, the eigenvalues satisfy the assumption*
$$\exists \alpha > 1, c > 0 : \lambda_i \le ci^{-\alpha},$$

*and the size of an uniform sampling is*
$$m \ge \Omega\left(\sqrt{n}\log(1/\delta)\right).$$

*then, with probability at least $1 - \delta$, we have*
$$\mathbb{E}[\mathcal{W}(\mathbf{C}_{n,m}, \mathbb{P})] - \mathcal{W}^*(\mathbb{P}) \le \mathcal{O}\left(\sqrt{\frac{k}{n}}\log\left(\frac{n}{\delta}\right)\right).$$

The assumption of algebraically decreasing eigenvalues of the kernel matrix is a common assumption, and met by the popular finite rank kernels and shift invariant kernel [54], for example. The above results show that we can guarantee the optimal generalization performance when only sampling $\tilde{\Omega}(\sqrt{n})$ points, which is much better than $\tilde{\Omega}(\sqrt{nk})$ when $k$ is large.

## 4.4 Further Results: Beyond ERM

In the following, we show that our result can be extended to general cases beyond ERM.

**Theorem 6.** *Under the same assumptions as Theorem 5, if*
$$\mathbb{E}\left[\mathcal{W}(\tilde{\mathbf{C}}_{n,m}, \mathbb{P}_n) - \mathcal{W}(\mathbf{C}_{n,m}, \mathbb{P}_n)\right] \le \zeta,$$

*and the size of an uniform sampling is*
$$m \ge \Omega\left(\sqrt{n}\log(1/\delta)\right),$$

*then, with probability at least $1 - \delta$, we have*
$$\mathbb{E}[\mathcal{W}(\tilde{\mathbf{C}}_{n,m}, \mathbb{P})] - \mathcal{W}^*(\mathbb{P}) \le \tilde{\mathcal{O}}\left(\sqrt{\frac{k}{n}} + \zeta\right).$$

The above result demonstrates that the risk bound of $\tilde{\mathbf{C}}_{n,m}$ is optimal when $\mathbb{E}\left[\mathcal{W}(\tilde{\mathbf{C}}_{n,m}, \mathbb{P}_n) - \mathcal{W}(\mathbf{C}_{n,m}, \mathbb{P}_n)\right]$ is small.

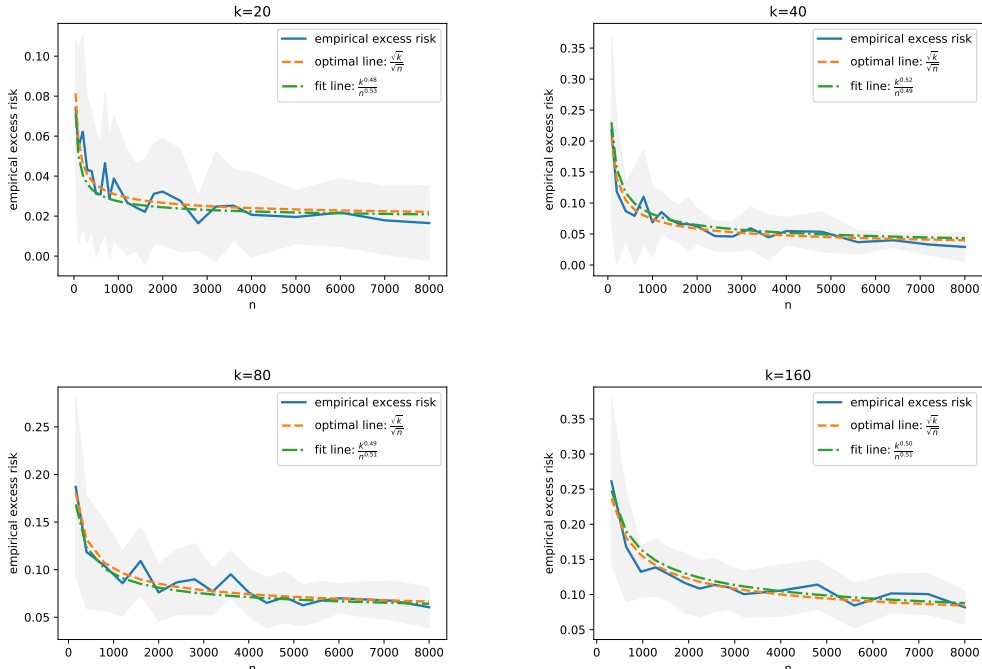

Figure 1: The empirical excess error of kernel $k$-means on the test set with different sizes of training data $n$ and the number of clustering $k$. The blue line means the empirical excess error on the test set with different sizes of training data. The dotted orange line means the optimal rate of theoretical findings. The dotted green line means the fit curve of the empirical excess error.

## 4.5 Further Results: $k$-means ++

If adopting the improved $k$-kernel means++ sampling with a local search strategy [25] for Nyström kernel $k$-means, we can obtain the following results:

**Theorem 7.** *Under the same assumptions as Theorem 5, $\mathbf{C}_{n,m}^{\mathcal{A}}$ is returned by the improved $k$-means++ algorithm with a local search strategy [25], if the size of an uniform sampling is*

$$m \geq \Omega\left(\sqrt{n}\log(1/\delta)\right),$$

*then with probability at least $1 - \delta$, we have*

$$\mathbb{E}_{\mathcal{D}}\left[\mathbb{E}_{\mathcal{A}}[\mathcal{W}(\mathbf{C}_{n,m}^{\mathcal{A}}, \mathbb{P})]\right] \leq \tilde{\mathcal{O}}\left(\sqrt{\frac{k}{n}} + \mathcal{W}^*(\mathbb{P})\right),$$

*where $\mathcal{A}$ is the randomness derived from the $k$-means++ initialization.*

The above result implies that if the optimal clustering risk $\mathcal{W}^*(\mathbb{P})$ is small, i.e. $\mathcal{W}^*(\mathbb{P}) \leq \tilde{\mathcal{O}}(\sqrt{k/n})$, the risk of $\mathcal{W}(\mathbf{C}_{n,m}^{\mathcal{A}}, \mathbb{P})$ can reach $\tilde{\mathcal{O}}(\sqrt{k/n})$.

## 5 Experiments

In this section, we will validate our theoretical findings by performing experiments on both simulated data and real data for kernel $k$-means and approximate $k$-means.

### 5.1 Numerical Experiments

In this subsection, we will validate our theoretical findings by performing experiments on simulated data for kernel $k$-means and approximate $k$-means.

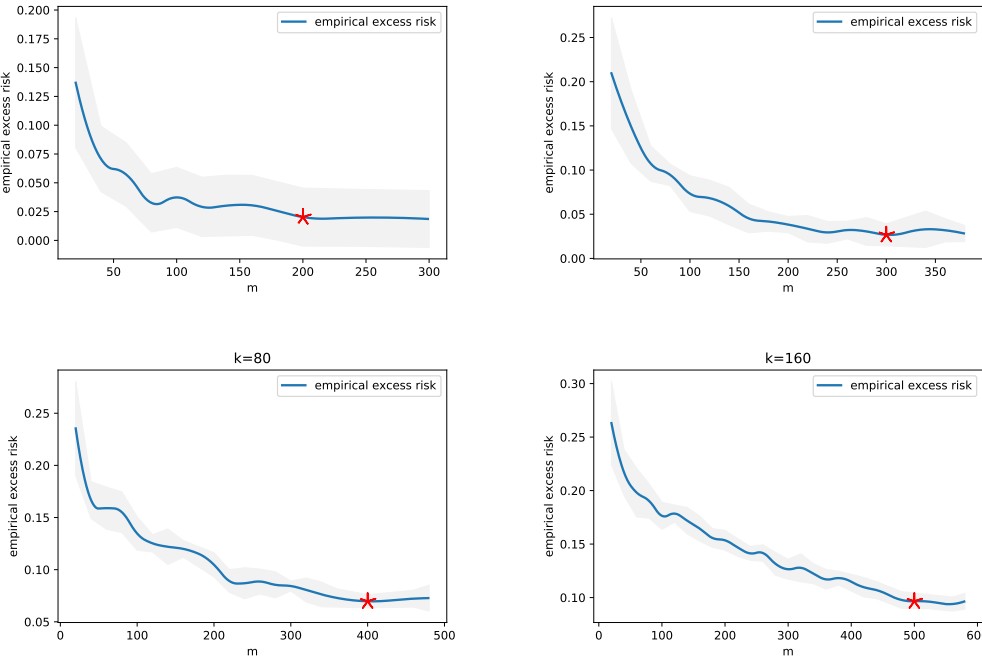

Figure 2: The empirical excess error of the approximate kernel $k$-means on the test set with different uniform samplings $m$. The red star is the lower bound of the sampling landmarks, which, when increased, does not decrease the error.

Let $\mathbf{c}_i^* \in \mathbb{R}^{10}$, $i = 1, \ldots, k$, be the clustering centers, where the values of the 10 dimensions are 1 or $-1$ with equal probability. We generate the $i$th clustering samples $\mathcal{C}_i$ from the normal distribution with mean $\mathbf{c}_i^*$ and variance 2, $|\mathcal{C}_1| = \ldots = |\mathcal{C}_k|$. In the experiments, we consider the popular Gaussian kernel

$$\kappa(\mathbf{x}, \mathbf{x}') = \exp\left(-\frac{\|\mathbf{x} - \mathbf{x}'\|^2}{10}\right).$$

Based on the above construction method, it is easy to verify that the optimal clustering risk is

$$\mathcal{W}^*(\mathbb{P}) = \int \min_{j=1,\ldots,k} \|\Phi_{\mathbf{x}} - \Phi_{\mathbf{c}_j^*}\|^2 d\mathbb{P}(\mathbf{x}) = \int \min_{j=1,\ldots,k} 2(1 - \kappa(\mathbf{x}, \mathbf{c}_j^*)) d\mathbb{P}(\mathbf{x}).$$

**Kernel $k$-Means**

In the first experiment, we validate our theoretical findings of kernel $k$-means. We generate $\sum_{i=1}^{k} |\mathcal{C}_i|$ samples of $k$ clustering centers for training and 10,000 samples for testing. The empirical excess risk of kernel $k$-means on the test set can be written as

$$\frac{\sum_{\mathbf{x}_i \in \mathcal{D}_t} \min_{j=1,\ldots,k} \|\Phi_{\mathbf{x}_i} - \Phi_{\mathbf{c}_j}\|^2 - \min_{j=1,\ldots,k} \|\Phi_{\mathbf{x}_i} - \Phi_{\mathbf{c}_j^*}\|^2}{|\mathcal{D}_t|},$$

where $\mathbf{C}_n = [\mathbf{c}_1, \ldots, \mathbf{c}_k]$ is the solution returned by the kernel $k$-means using Lloyd's algorithm [40], and $\mathcal{D}_t$ is the test set.

The empirical excess errors of kernel $k$-means on the test set with different sizes of training data and numbers of $k$ are given in Figure 1. We can see that the line of best fit for empirical excess risks is $\frac{k^{0.48}}{n^{0.53}}$ for $k = 20$, $\frac{k^{0.52}}{n^{0.49}}$ for $k = 40$, $\frac{k^{0.49}}{n^{0.51}}$ for $k = 80$, and $\frac{k^{0.50}}{n^{0.51}}$ for $k = 160$, achieving the predicted rate $\frac{k^{0.5}}{n^{0.5}}$ (from Theorem 1), which verifies our theoretical findings.

**Approximate Kernel $k$-Means**

In the second experiment, we validate our theoretical findings of approximate kernel $k$-means on simulated data.

The data generation rule is the same as that in the kernel $k$-means. We generate 10,000 samples ($|\mathcal{C}_i| = 10000/k$) for training and 10,000 samples for testing. The empirical excess errors of the approximate kernel $k$-means on the test set with different uniform samplings $m$ are given in Figure 2, which can be summarized as follows:

1) There exists a lower bound of the sampling landmarks l which does not decrease the error when increase its value. This verifies the theoretical statement in Theorem 4.

2) The lower bound of $l$ increases with the number of the clusters $k$. This result confirms Theorem 4 once again.

## 5.2 Real-World Scenarios

To reflect real-world scenarios, we add more experiments on the real data sets. We use 6 publicly avaiable datasets, dna, segment, mushrooms, mnist, skin-nonskin and covtype, from the LIBSVM Data [2]. The empirical evaluations with Gaussian kernel

$$\exp\left(-\frac{\|\mathbf{x} - \mathbf{x}'\|^2}{\sigma^2}\right), \sigma = \sqrt{\frac{\sum_{ij}\|x_i - x_j\|^2}{d}},$$

are given in the following table 1, where $d$ is the dimension of $\mathbf{x} \in \mathcal{X}$.

Table 1: Experiments on the real data sets with kernel $k$-means and NyStöm kernel $k$-means ($m = 100$).

| Dataset | Datasize | Kernel $k$-means | Nyström Kernel $k$-Means |
|---------|----------|------------------|--------------------------|
| dna | 2000 | 0.53 | 0.52 |
| segment | 2310 | 0.55 | 0.55 |
| mushrooms | 8124 | 0.66 | 0.65 |
| mnist | 60000 | – | 0.43 |
| skin-nonskin | 245057 | – | 0.63 |
| covtype | 581012 | – | 0.32 |

From the above results on real data sets, we can find that Nyström kernel $k$-means give the similar results as the original one, which also match the theoretical findings.

## 6 Conclusion

In this paper, we derive nearly optimal risk bounds for both kernel $k$-means and Nyström kernel $k$-means of learning rate of $\mathcal{O}(\sqrt{k/n})$, which fills the gap ignoring the optimal risk bounds for (approximate) kernel $k$-means. Furthermore, we extend these results to general cases beyond ERM and $k$-means++. Our result may provide a new perspective to study the optimal statistical properties of unsupervised learning.

In this paper we only derived the risk bounds of learning rate $\mathcal{O}(\sqrt{k/n})$ for the basic case. In the future, we will consider studying whether it is possible to prove a bound of $\mathcal{O}(\sqrt{k}/n)$ under certain strict assumptions.

## Acknowledgment

This work is supported in part by the National Natural Science Foundation of China (No. 62076234, No.61703396, No.62106257), Beijing Outstanding Young Scientist Program NO.BJJWZYJH012019100020098, Intelligent Social Governance Platform, Major Innovation & Planning Interdisciplinary Platform for the "Double-First Class" initiative, Renmin University of China, China Unicom Innovation Ecological Cooperation Plan, Public Computing Cloud of Renmin University of China.

---

[2]http://www.csie.ntu.edu.tw/ cjlin/libsvm

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
