# Supplementary Material for "Refined Learning Bounds for Kernel and Approximate $k$-Means"

**Yong Liu**[1,2]
[1]Gaoling School of Artificial Intelligence, Renmin University of China, Beijing, China
[2]Beijing Key Laboratory of Big Data Management and Analysis Methods, Beijing, China
liuyonggsai@ruc.edu.cn

## Appendix: Upper Bound for the Clustering Rademacher Complexity

Let $\mathcal{F}_{\mathbf{C}}$ be a family of $k$-*valued* functions with

$$\mathcal{F}_{\mathbf{C}} := \Big\{ f_{\mathbf{C}} = (f_{\mathbf{c}_1}, \ldots, f_{\mathbf{c}_k}) : \mathbf{C} \in \mathcal{H}^k \Big\}. \tag{1}$$

Let $\varphi : \mathbb{R}^k \to \mathbb{R}$ be a minimum function:

$$\forall \boldsymbol{\alpha} \in \mathbb{R}^k, \varphi(\boldsymbol{\alpha}) = \min_{i=1,\ldots,k} \alpha_i \tag{2}$$

and $\mathcal{G}_{\mathbf{C}}$ be a "minimum" family of the functions $\mathcal{F}_{\mathbf{C}}$,

$$\mathcal{G}_{\mathbf{C}} := \Big\{ g_{\mathbf{C}} = \varphi \circ f_{\mathbf{C}} \ \Big| \ f_{\mathbf{C}} \in \mathcal{F}_{\mathbf{C}}, g_{\mathbf{C}}(\mathbf{x}) = \varphi(f_{\mathbf{C}}(\mathbf{x})) \Big\}. \tag{3}$$

**Definition 1** (Clustering Rademacher Complexity). *Let $\mathcal{G}_{\mathbf{C}}$ be a family of functions defined in* (3), $\mathcal{S} = (\mathbf{x}_1, \ldots, \mathbf{x}_n)$ *be a fixed sample of size $n$ with elements in $\mathcal{X}$, and $\mathcal{D} = \{\Phi_i = \psi(\mathbf{x}_i)\}_{i=1}^n$. Then, the clustering empirical Rademacher complexity of $\mathcal{G}_{\mathbf{C}}$ with respect to $\mathcal{D}$ is defined by*

$$\mathcal{R}_n(\mathcal{G}_{\mathbf{C}}) = \mathbb{E}_{\boldsymbol{\sigma}} \left[ \sup_{g_{\mathbf{C}} \in \mathcal{G}_{\mathbf{C}}} \left| \sum_{i=1}^n \sigma_i g_{\mathbf{C}}(\mathbf{x}_i) \right| \right],$$

*where $\sigma_1, \ldots, \sigma_n$ are independent random variables with equal probability of taking values $+1$ or $-1$. Its expectation is $\mathcal{R}(\mathcal{G}_{\mathbf{C}}) = \mathbb{E}\left[\mathcal{R}_n(\mathcal{G}_{\mathbf{C}})\right].$*

Based on the recently improvement of the upper bound of Rademacher complexity of $L$-Lipschitz with respect to the $L_\infty$ norm [5], we provide a refined bound of clustering Rademacher complexity:

**Lemma 1.** *If $\forall \mathbf{x} \in \mathcal{X}, \|\Phi_{\mathbf{x}}\| \leq 1$, then, for any $\mathcal{S} = \{\mathbf{x}_1, \ldots, \mathbf{x}_n\} \in \mathcal{X}^n$, there exists a constant $c > 0$ such that*

$$\mathcal{R}_n(\mathcal{G}_{\mathbf{C}}) \leq c\sqrt{k} \max_i \tilde{\mathcal{R}}_n(\mathcal{F}_{\mathbf{C}_i}) \log^2(\sqrt{n}),$$

*where $\mathcal{G}_{\mathbf{C}}$ is a family of clustering functions defined in* (3), $\mathcal{F}_{\mathbf{C}}$ *is a family of $k$-valued functions associate with the clustering center $\mathbf{C} = [\mathbf{c}_1, \ldots, \mathbf{c}_k]$ defined in* (1), $\mathcal{F}_{\mathbf{C}_i}$ *is a family of the output coordinate $i$ of $\mathcal{F}_{\mathbf{C}}$, and $\tilde{\mathcal{R}}_n(\mathcal{F}_{\mathbf{C}_i}) = \sup_{\mathcal{S} \in \mathcal{X}^n} \mathcal{R}_n(\mathcal{F}_{\mathbf{C}_i})$.*

The above result shows that the upper bound of the clustering Rademacher complexity is linearly dependent on $\sqrt{k}$, which substantially improves the existing bounds linearly dependent on $k$.

**Remark.** The upper bound of the clustering Rademacher complexity involves a constant $c$ and a logarithmic term $\log(n)$. Thus, if one requires its absolute value to be smaller than the existing bounds defined, there may exist some cases which acquire a large $k$. However, from a statistical perspective, our bound with linear dependence on $\sqrt{k}$ substantially improves the existing ones with linear dependence on $k$.

In the following, we will show that Lemma 1 cannot be improved from a statistical view when ignoring the logarithmic terms.

35th Conference on Neural Information Processing Systems (NeurIPS 2021).

**Lemma 2.** *There exists a set $\mathbf{C} \in \mathcal{H}^k$ and data sequence $\mathcal{D} = \{\Phi_1, \ldots, \Phi_n\}$ such that*

$$\mathcal{R}_n(\mathcal{G}_\mathbf{C}) \geq \frac{\sqrt{k}}{3\sqrt{2}} \cdot \max_i \tilde{\mathcal{R}}_n(\mathcal{F}_{\mathbf{C}_i}).$$

Lemma 2 shows that the lower bound of $\mathcal{R}_n(\mathcal{G}_\mathbf{C})$ is $\Omega\big(\sqrt{k}\max_i \tilde{\mathcal{R}}_n(\mathcal{F}_{\mathbf{C}_i})\big)$, which implies that the upper bound of order $\tilde{\mathcal{O}}\big(\sqrt{k}\max_i \tilde{\mathcal{R}}_n(\mathcal{F}_{\mathbf{C}_i})\big)$ in Lemma 1 is **(nearly) optimal** when ignoring the logarithmic terms

**Remark.** A lower bound linearly dependent on $k$ for a $k$-valued function class $\mathcal{F} \subseteq \{f : \mathcal{X} \to \mathbb{R}^k\}$ has been given in [5],

$$\mathcal{R}_n(\phi \circ \mathcal{F}) \geq \frac{k}{2\sqrt{2}} \cdot \max_i \tilde{\mathcal{R}}_n(\phi \circ \mathcal{F}_i),$$

which does not match the upper bound of $\sqrt{k}$. However our bound in Lemma 2 does match.

## Appendix: Proof of Lemma 1

To prove Lemma 1, we first give the following two lemmas:

**Lemma 3** ($L_\infty$ *Contraction Inequality, Theorem 1 in [5]*). *Let $\mathcal{F} \subseteq \{f : \mathcal{X} \to \mathbb{R}^k\}$, and let $\phi : \mathbb{R}^k \to \mathbb{R}$ be $L$-Lipschitz with respect to the $L_\infty$ norm, that is $\|\phi(\mathbf{v}) - \phi(\mathbf{v}')\|_\infty \leq L \cdot \|\mathbf{v} - \mathbf{v}'\|_\infty$, $\forall \mathbf{v}, \mathbf{v}' \in \mathbb{R}^k$. For any $a > 0$, there exists a constant $C > 0$ such that if $\max\{|\phi(f(\mathbf{x}))|, \|f(\mathbf{x})\|_\infty\} \leq \rho$, then*

$$\mathcal{R}_n(\phi \circ \mathcal{F}) \leq C \cdot L\sqrt{k}\max_i \tilde{\mathcal{R}}_n(\mathcal{F}_i) \log^{\frac{3}{2}+a}\left(\frac{\rho n}{\max_i \tilde{\mathcal{R}}_n(\mathcal{F}_i)}\right),$$

*where $\mathcal{R}_n(\phi \circ \mathcal{F}) = \mathbb{E}_{\boldsymbol{\sigma}}\left[\sup_{f \in \mathcal{F}} |\sum_{i=1}^n \sigma_i \phi(f(\mathbf{x}_i))|\right]$, $\tilde{\mathcal{R}}_n(\mathcal{F}_i) = \sup_{\mathcal{S} \in \mathcal{X}^n} \mathcal{R}_n(\mathcal{F}_i)$.*

**Lemma 4** (*Lemma 24(a) in [7] with $p = 2$*). *Let $\eta_1, \ldots, \eta_n \in \mathcal{H}$, where $\mathcal{H}$ is a Hilbert space with $\|\cdot\|$ being the associated norm. Let $\sigma_1, \ldots, \sigma_n$ be a sequence of independent Rademacher variables. Then, we have*

$$\mathbb{E}_{\boldsymbol{\sigma}}\left\|\sum_{i=1}^n \sigma_i \eta_i\right\|^2 \leq \sum_{i=1}^n \|\eta_i\|^2 \tag{4}$$

*and*

$$\mathbb{E}\left\|\sum_{i=1}^n \sigma_i \eta_i\right\| \geq \frac{\sqrt{2}}{2}\sqrt{\sum_{i=1}^n \|\eta_i\|^2}. \tag{5}$$

*Proof of Lemma 1.* We first show that the minimum function

$$\varphi(\boldsymbol{\nu}) = \min(\nu_1, \ldots, \nu_k)$$

defined in (2) is 1-Lipschitz continuous with respect to the $L_\infty$-norm, that is

$$\forall \boldsymbol{\nu}, \boldsymbol{\nu}' \in \mathbb{R}^k, |\varphi(\boldsymbol{\nu}) - \varphi(\boldsymbol{\nu}')| \leq \|\boldsymbol{\nu} - \boldsymbol{\nu}'\|_\infty. \tag{6}$$

Without loss of generality, we assume that $\varphi(\boldsymbol{\nu}) \geq \varphi(\boldsymbol{\nu}')$. Let

$$j = \underset{i=1,\ldots,k}{\arg\min} \nu_i',$$

then from the definition of $\varphi$, we know that $\varphi(\boldsymbol{\nu}') = \nu_j'$. Thus, we can obtain that

$$\begin{aligned}
|\varphi(\boldsymbol{\nu}) - \varphi(\boldsymbol{\nu}')| &= \varphi(\boldsymbol{\nu}) - \nu_j' \\
&\leq \nu_j - \nu_j' \qquad\qquad \text{(by the fact that } \varphi(\boldsymbol{\nu}) \leq \nu_j) \\
&\leq \|\boldsymbol{\nu} - \boldsymbol{\nu}'\|_\infty.
\end{aligned}$$

We then show that $\max\{|\varphi(f_{\mathbf{C}}(\mathbf{x}))|, \|f_{\mathbf{C}}(\mathbf{x})\|_\infty\}$ is bounded by a constant. From the definition of $f_{\mathbf{C}}$ (see Eq.(1)), we know that

$$f_{\mathbf{C}}(\mathbf{x}) = (f_{\mathbf{c}_1}(\mathbf{x})\ldots, f_{\mathbf{c}_k}(\mathbf{x})) \text{ and } f_{\mathbf{c}_j}(\mathbf{x}) = \|\Phi_{\mathbf{x}} - \mathbf{c}_j\|^2.$$

Note that $\|\Phi_{\mathbf{x}}\| \leq 1$ and $\mathbf{c}_j \in \mathcal{H}$, so we have

$$\|\mathbf{c}_j\| \leq 1 \text{ and } f_{\mathbf{c}_j}(\mathbf{x}) \leq 2\|\Phi_{\mathbf{x}}\| + 2\|\mathbf{c}_j\| \leq 4, \forall \mathbf{x} \in \mathcal{X}. \tag{7}$$

Thus, one can see that

$$\|f_{\mathbf{C}}(\mathbf{x})\|_\infty = \max_j |f_{\mathbf{c}_j}(\mathbf{x})| \leq 4 \text{ and } |\varphi(f_{\mathbf{C}}(\mathbf{x}))| = |\min_{j=1,\ldots,k} f_{\mathbf{c}_j}(\mathbf{x})| \leq 4.$$

From the above analysis, we know that $\varphi(\boldsymbol{\nu})$ is 1-continuous with respect to the $L_\infty$-norm, and $\max\{|\varphi(f_{\mathbf{C}}(\mathbf{x}))|, \|f_{\mathbf{C}}(\mathbf{x})\|_\infty\} \leq 4$. Thus, using Lemma 3 with $L = 1$, $\rho = 4$ and $a = 1/2$, we have

$$\mathcal{R}_n(\mathcal{G}_{\mathbf{C}}) \leq C\sqrt{k} \max_i \tilde{\mathcal{R}}_n(\mathcal{F}_{\mathbf{C}_i}) \log^2\left(\frac{4n}{\max_i \tilde{\mathcal{R}}_n(\mathcal{F}_{\mathbf{C}_i})}\right). \tag{8}$$

Let

$$c_i := \sup_{\mathbf{x} \in \mathcal{X}} \sup_{f_{\mathbf{c}} \in \mathcal{F}_{\mathbf{C}_i}} |f_{\mathbf{c}}(\mathbf{x})| \text{ and } c = \max\{c_i, i = 1, \ldots, k\}. \tag{9}$$

From (7), we know that $c$ is a constant and $c \leq 4$. By definition of $\tilde{\mathcal{R}}_n(\mathcal{F}_{\mathbf{C}_i})$, we can obtain that

$$
\begin{aligned}
\forall j, \tilde{\mathcal{R}}_n(\mathcal{F}_{\mathbf{C}_j}) &= \sup_{S \in \mathcal{X}^n} \mathbb{E}_{\boldsymbol{\sigma}} \left[ \sup_{f_{\mathbf{c}} \in \mathcal{F}_{\mathbf{C}_j}} \left| \sum_{i=1}^n \sigma_i f_{\mathbf{c}}(\mathbf{x}_i) \right| \right] \\
&\geq \sup_{\mathbf{x} \in \mathcal{X}} \mathbb{E}_{\boldsymbol{\sigma}} \left[ \sup_{f_{\mathbf{c}} \in \mathcal{F}_{\mathbf{C}_j}} \left| \sum_{i=1}^n \sigma_i f_{\mathbf{c}}(\mathbf{x}) \right| \right] \\
&\geq \sup_{\mathbf{x} \in \mathcal{X}, f_{\mathbf{c}} \in \mathcal{F}_{\mathbf{C}_j}} \mathbb{E}_{\boldsymbol{\sigma}} \left| \sum_{i=1}^n \sigma_i f_{\mathbf{c}}(\mathbf{x}) \right| \quad \text{(by Jensen's inequality)} \\
&\geq \frac{\sqrt{2n}}{2} \sup_{\mathbf{x} \in \mathcal{X}, f_{\mathbf{c}} \in \mathcal{F}_{\mathbf{C}_j}} \sqrt{|f_{\mathbf{c}}(\mathbf{x})|} \quad \text{(by Eq.(5) of Lemma 4)} \\
&= \frac{\sqrt{2nc_j}}{2} \quad \text{(by Eq.(9)).}
\end{aligned}
\tag{10}
$$

Thus, one can see that $\max_i \tilde{\mathcal{R}}_n(\mathcal{F}_{\mathbf{C}_i}) \geq \frac{\sqrt{2cn}}{2}$, where $c = \max\{c_i, i = 1, \ldots, k\}$. So, we have $\frac{n}{\max_i \tilde{\mathcal{R}}_n(\mathcal{F}_{\mathbf{C}_i})} \leq \sqrt{\frac{2n}{c}}$. Plugging this into (8) proves the result. $\square$

## Appendix: Proof of Theorem 1

To prove Theorem 1, we first give the following two lemmas:

**Lemma 5.** *If $\forall \mathbf{x} \in \mathcal{X}, \|\Phi_{\mathbf{x}}\| \leq 1$, then for all $S \in \mathcal{X}^n$ and $\mathbf{C} \in \mathcal{H}^k$, we have*

$$\max_i \tilde{\mathcal{R}}_n(\mathcal{F}_{\mathbf{C}_i}) \leq 3\sqrt{n}.$$

*Proof.* $\forall \mathcal{S} \in \mathcal{X}^n$, $\mathbf{C} \in \mathcal{H}^k$ and $i \in \{1, \ldots, k\}$, we have

$$
\begin{aligned}
\mathcal{R}_n(\mathcal{F}_{\mathbf{C}_i}) &= \mathbb{E}_{\boldsymbol{\sigma}} \sup_{f_{\mathbf{c}} \in \mathcal{F}_{\mathbf{C}_i}} \left| \sum_{j=1}^n \sigma_j f_{\mathbf{c}}(\mathbf{x}_j) \right| \\
&= \mathbb{E}_{\boldsymbol{\sigma}} \sup_{\mathbf{c} \in \mathcal{H}} \left| \sum_{j=1}^n \sigma_j \|\Phi_j - \mathbf{c}\|^2 \right| \\
&= \mathbb{E}_{\boldsymbol{\sigma}} \sup_{\mathbf{c} \in \mathcal{H}} \left| \sum_{j=1}^n \sigma_j \left[ -2\langle \Phi_j, \mathbf{c} \rangle + \|\mathbf{c}\|^2 + \|\Phi_j\|^2 \right] \right| \\
&= \mathbb{E}_{\boldsymbol{\sigma}} \sup_{\mathbf{c} \in \mathcal{H}} \left| \sum_{j=1}^n \sigma_j \left[ -2\langle \Phi_j, \mathbf{c} \rangle + \|\mathbf{c}\|^2 \right] \right| \\
&\leq 2\mathbb{E}_{\boldsymbol{\sigma}} \sup_{\mathbf{c} \in \mathcal{H}} \left| \sum_{j=1}^n \sigma_j \langle \Phi_j, \mathbf{c} \rangle \right| + \mathbb{E}_{\boldsymbol{\sigma}} \sup_{\mathbf{c} \in \mathcal{H}} \left| \sum_{j=1}^n \sigma_j \|\mathbf{c}\|^2 \right|.
\end{aligned}
\tag{11}
$$

One can see that

$$
\begin{aligned}
\mathbb{E}_{\boldsymbol{\sigma}} \sup_{\mathbf{c} \in \mathcal{H}} \left| \sum_{j=1}^n \sigma_j \|\mathbf{c}\|^2 \right| &\leq \mathbb{E}_{\boldsymbol{\sigma}} \left| \sum_{j=1}^n \sigma_j \right| \quad \text{(since } \|\mathbf{c}\| \leq 1) \\
&\leq \sqrt{ \mathbb{E}_{\boldsymbol{\sigma}} \left| \sum_{j=1}^n \sigma_j \right|^2 } \leq \sqrt{n} \quad \text{(by Eq.(4) of Lemma 4)},
\end{aligned}
\tag{12}
$$

and

$$
\begin{aligned}
\mathbb{E}_{\boldsymbol{\sigma}} \sup_{\mathbf{c} \in \mathcal{H}} \left| \sum_{j=1}^n \sigma_j \langle \Phi_j, \mathbf{c} \rangle \right| &= \mathbb{E}_{\boldsymbol{\sigma}} \sup_{\mathbf{c} \in \mathcal{H}} \left| \left\langle \sum_{j=1}^n \sigma_j \Phi_j, \mathbf{c} \right\rangle \right| \\
&\leq \mathbb{E}_{\boldsymbol{\sigma}} \left\| \sum_{j=1}^n \sigma_j \Phi_j \right\| \quad \text{(by } \|\mathbf{c}\| \leq 1) \\
&\leq \sqrt{ \mathbb{E}_{\boldsymbol{\sigma}} \left\| \sum_{j=1}^n \sigma_j \Phi_j \right\|^2 } \leq \sqrt{ \sum_{i=1}^n \|\Phi_i\|^2 } \quad \text{(by Eq.(4) of Lemma 4)} \\
&\leq \sqrt{n} \quad \text{(since } \|\Phi_i\| \leq 1).
\end{aligned}
\tag{13}
$$

Substituting (12) and (13) into (11), we can prove the result. $\qquad \square$

To prove Theorem 1, we first propose the following lemma:

**Lemma 6.** *For any $\delta \in (0, 1)$, with probability $1 - \delta$, there exists a constant $c > 0$, such that*

$$
\mathcal{R}(\mathcal{G}_{\mathbf{C}}) \leq c\sqrt{kn} \log^2\left(\sqrt{n}\right) + \sqrt{2n \log\left(\frac{1}{\delta}\right)}.
$$

*Proof.* From [8] or [1], with probability $1 - \delta$, we have

$$
\mathcal{R}(\mathcal{G}_{\mathbf{C}}) \leq \mathcal{R}_n(\mathcal{G}_{\mathbf{C}}) + \sqrt{2n \log\left(\frac{1}{\delta}\right)}.
\tag{14}
$$

Thus, we have

$$
\mathcal{R}(\mathcal{G}_{\mathbf{C}})
$$

$$
\leq \mathcal{R}_n(\mathcal{G}_{\mathbf{C}}) + \sqrt{2n \log\left(\frac{1}{\delta}\right)}
$$

$$
\leq c\sqrt{k} \max_i \tilde{\mathcal{R}}_n(\mathcal{F}_{\mathbf{C}_i}) \log^2\left(\sqrt{n}\right) + \sqrt{2n \log\left(\frac{1}{\delta}\right)} \qquad \text{(by Lemma 1)}
$$

$$
\leq 3c\sqrt{kn} \log^2\left(\sqrt{n}\right) + \sqrt{2n \log\left(\frac{1}{\delta}\right)}. \qquad \text{(by Lemma 5)}
$$

$\square$

*Proof of Theorem 1.* The starting point of our analysis is the following elementary inequality (see Ch.8 in [4] or page 2 in [3]):

$$
\begin{aligned}
&\mathbb{E}[\mathcal{W}(\mathbf{C}_n, \mathbb{P})] - \mathcal{W}^*(\mathbb{P}) \\
=&\mathbb{E}\left[\mathcal{W}(\mathbf{C}_n, \mathbb{P}) - \mathcal{W}(\mathbf{C}_n, \mathbb{P}_n)\right] + \mathbb{E}\left[\mathcal{W}(\mathbf{C}_n, \mathbb{P}_n)\right] - \mathcal{W}^*(\mathbb{P}) \\
\leq&\mathbb{E}\left[\mathcal{W}(\mathbf{C}_n, \mathbb{P}) - \mathcal{W}(\mathbf{C}_n, \mathbb{P}_n)\right] + \mathbb{E}\left[\mathcal{W}(\mathbf{C}^*, \mathbb{P}_n)\right] - \mathcal{W}^*(\mathbb{P}) \\
&(\mathcal{W}(\mathbf{C}_n, \mathbb{P}_n) \leq \mathcal{W}(\mathbf{C}^*, \mathbb{P}_n) \text{ as } \mathbf{C}_n \text{ is optimal w.r.t. } \mathcal{W}(\cdot, \mathbb{P}_n)) \\
\leq&\mathbb{E} \sup_{\mathbf{C}\in\mathcal{H}^k} \left(\mathcal{W}(\mathbf{C}, \mathbb{P}) - \mathcal{W}(\mathbf{C}, \mathbb{P}_n)\right) + \sup_{\mathbf{C}\in\mathcal{H}^k} \mathbb{E}\left[\mathcal{W}(\mathbf{C}, \mathbb{P}_n) - \mathcal{W}(\mathbf{C}, \mathbb{P})\right] \\
\leq&2\mathbb{E} \sup_{\mathbf{C}\in\mathcal{H}^k} \left|\mathcal{W}(\mathbf{C}, \mathbb{P}_n) - \mathcal{W}(\mathbf{C}, \mathbb{P})\right|.
\end{aligned} \tag{15}
$$

Let $\mathbf{x}'_1, \ldots, \mathbf{x}'_n$ be a copy of $\mathbf{x}_1, \ldots, \mathbf{x}_n$, independent of the $\sigma_i$'s. Then, by a standard symmetrization argument [1] (can also be seen in the proof of Lemma 4.3 of [3]), we can write

$$
\begin{aligned}
\mathbb{E} \sup_{\mathbf{C}\in\mathcal{H}^k} \left|\mathcal{W}(\mathbf{C}, \mathbb{P}_n) - \mathcal{W}(\mathbf{C}, \mathbb{P})\right| \leq&\mathbb{E} \sup_{g_{\mathbf{C}}\in\mathcal{G}_{\mathbf{C}}} \left|\frac{1}{n} \sum_{i=1}^n \sigma_i \left[g_{\mathbf{C}}(\mathbf{x}) - g_{\mathbf{C}}(\mathbf{x}')\right]\right| \\
\leq&2\mathbb{E} \sup_{g_{\mathbf{C}}\in\mathcal{G}_{\mathbf{C}}} \left|\frac{1}{n} \sum_{i=1}^n \sigma_i g_{\mathbf{C}}(\mathbf{x})\right| = \frac{2}{n}\mathcal{R}(\mathcal{G}_{\mathbf{C}}).
\end{aligned} \tag{16}
$$

Thus, we can obtain that

$$
\mathbb{E}\left[\mathcal{W}(\mathbf{C}_n, \mathbb{P})\right] - \mathcal{W}^*(\mathbb{P}) \leq \frac{4}{n}\mathcal{R}(\mathcal{G}_{\mathbf{C}}) \text{ (by Eq.(15) and Eq.(16))}
$$

$$
\leq 4c\sqrt{\frac{k}{n}} \log^2\left(\sqrt{n}\right) + 4\sqrt{\frac{2\log\frac{1}{\delta}}{n}} \qquad \text{(by Lemma 6).}
$$

This proves the result. $\square$

## Appendix: Proof of Theorem 2

*Proof.* Note that

$$
\begin{aligned}
&\mathbb{E}[\mathcal{W}(\tilde{\mathbf{C}}_n, \mathbb{P})] - \mathcal{W}^*(\mathbb{P}) \\
=&\underbrace{\mathbb{E}\left[\mathcal{W}(\tilde{\mathbf{C}}_n, \mathbb{P}) - \mathcal{W}(\tilde{\mathbf{C}}_n, \mathbb{P}_n)\right]}_{A_1} + \underbrace{\mathbb{E}\left[\mathcal{W}(\tilde{\mathbf{C}}_n, \mathbb{P}_n) - \mathcal{W}(\mathbf{C}_n, \mathbb{P}_n)\right]}_{A_2} \\
&+ \underbrace{\mathbb{E}\left[\mathcal{W}(\mathbf{C}_n, \mathbb{P}_n) - \mathcal{W}(\mathbf{C}_n, \mathbb{P})\right]}_{A_3} + \underbrace{\mathbb{E}\left[\mathcal{W}(\mathbf{C}_n, \mathbb{P})\right] - \mathcal{W}^*(\mathbb{P})}_{A_4}.
\end{aligned}
$$

Also note that $A_2$ is bounded by $\zeta$, and $A_4$ can be obtained from Theorem 1. From Eq.(16), we know that $A_1$ and $A_3$ can be bounded by the Rademacher complexity:

$$A_1 \leq \mathbb{E} \sup_{\mathbf{C} \in \mathcal{H}^k} |\mathcal{W}(\mathbf{C}, \mathbb{P}_n) - \mathcal{W}(\mathbf{C}, \mathbb{P})| \leq \frac{2}{n} \mathcal{R}(\mathcal{G}_\mathbf{C}),$$

$$A_3 \leq \mathbb{E} \sup_{\mathbf{C} \in \mathcal{H}^k} |\mathcal{W}(\mathbf{C}, \mathbb{P}_n) - \mathcal{W}(\mathbf{C}, \mathbb{P})| \leq \frac{2}{n} \mathcal{R}(\mathcal{G}_\mathbf{C}).$$

Thus, we can obtain that

$$\mathbb{E}[\mathcal{W}(\tilde{\mathbf{C}}_n, \mathbb{P})] - \mathcal{W}^*(\mathbb{P}) \leq \frac{4}{n} \mathcal{R}(\mathcal{G}_\mathbf{C}) + c\sqrt{\frac{k}{n}} \log^2\left(\sqrt{n}\right) + c\sqrt{\frac{\log \frac{1}{\delta}}{n}} + \zeta. \tag{17}$$

Substituting Lemma 6 into Eq.(17), we can proves the result. $\qquad \square$

## Appendix: Proof of Theorem 3

*Proof.* Note that

$$\mathbb{E}\left[\mathbb{E}_\mathcal{A}[\mathcal{W}(\mathbf{C}_n^\mathcal{A}, \mathbb{P})]\right] = \mathbb{E}\left[\mathbb{E}_\mathcal{A}[\mathcal{W}(\mathbf{C}_n^\mathcal{A}, \mathbb{P})] - \mathbb{E}_\mathcal{A}[\mathcal{W}(\mathbf{C}_n^\mathcal{A}, \mathbb{P}_n)]\right] + \mathbb{E}\left[\mathbb{E}_\mathcal{A}[\mathcal{W}(\mathbf{C}_n^\mathcal{A}, \mathbb{P}_n)]\right].$$

From Lemma 2, we can obtain that

$$\mathbb{E}\left[\mathbb{E}_\mathcal{A}[\mathcal{W}(\mathbf{C}_n^\mathcal{A}, \mathbb{P}_n)]\right] \leq \beta \cdot \mathbb{E}[\mathcal{W}(\mathbf{C}_n, \mathbb{P}_n)]$$

$$= \beta \cdot \mathbb{E}\left[\mathcal{W}(\mathbf{C}_n, \mathbb{P}_n) - \mathcal{W}(\mathbf{C}_n, \mathbb{P})\right] + \beta \cdot \mathbb{E}\left[\mathcal{W}(\mathbf{C}_n, \mathbb{P})\right].$$

Thus, we can obtain that

$$\mathbb{E}\left[\mathbb{E}_\mathcal{A}[\mathcal{W}(\mathbf{C}_n^\mathcal{A}, \mathbb{P})]\right] \leq \underbrace{\mathbb{E}\left[\mathbb{E}_\mathcal{A}[\mathcal{W}(\mathbf{C}_n^\mathcal{A}, \mathbb{P})] - \mathbb{E}_\mathcal{A}[\mathcal{W}(\mathbf{C}_n^\mathcal{A}, \mathbb{P}_n)]\right]}_{A_1}$$

$$+ \beta \cdot \underbrace{\mathbb{E}\left[\mathcal{W}(\mathbf{C}_n, \mathbb{P}_n) - \mathcal{W}(\mathbf{C}_n, \mathbb{P})\right]}_{A_2} + \beta \cdot \underbrace{\mathbb{E}\left[\mathcal{W}(\mathbf{C}_n, \mathbb{P})\right]}_{A_3}. \tag{18}$$

Note that

$$A_1, A_2 \leq \mathbb{E} \sup_{\mathbf{C} \in \mathcal{H}^k} \left|\mathcal{W}(\mathbf{C}, \mathbb{P}_n) - \mathcal{W}(\mathbf{C}, \mathbb{P})\right|$$

$$\leq \frac{2}{n} \mathcal{R}(\mathcal{G}_\mathbf{C}) \qquad\qquad \text{(by Eq.(16))} \tag{19}$$

$$\leq \tilde{\mathcal{O}}\left(\sqrt{\frac{k}{n}}\right). \qquad\qquad \text{(by Lemma 6)}$$

By Theorem 1, we can obtain that

$$\mathbb{E}[\mathcal{W}(\mathbf{C}_n, \mathbb{P})] \leq \mathcal{W}^*(\mathbb{P}) + c\sqrt{\frac{k}{n}} \log^2\left(\sqrt{n}\right) + c\sqrt{\frac{\log \frac{1}{\delta}}{n}}.$$

Substituting the above inequality and Eq.(19) into Eq.(18), we have

$$\mathbb{E}\left[\mathbb{E}_\mathcal{A}[\mathcal{W}(\mathbf{C}_n^\mathcal{A}, \mathbb{P}_n)]\right] \leq \tilde{\mathcal{O}}\left(\sqrt{\frac{k}{n}} + \mathcal{W}^*(\mathbb{P})\right).$$

$\qquad \square$

# Appendix: Proof of Theorem 4

To prove Theorem 4, we first propose the following lemma:

**Lemma 7.** *With probability at least $1 - \delta$, we have*

$$\mathbb{E}\left[\mathcal{W}(\mathbf{C}_{n,m}, \mathbb{P}_n) - \mathcal{W}(\mathbf{C}_{n,m}, \mathbb{P})\right] \leq \tilde{\mathcal{O}}\left(\sqrt{\frac{k}{n}}\right).$$

*Proof.* Note that

$$
\begin{aligned}
\mathbb{E}\left[\mathcal{W}(\mathbf{C}_{n,m}, \mathbb{P}_n) - \mathcal{W}(\mathbf{C}_{n,m}, \mathbb{P})\right] \leq & \mathbb{E} \sup_{\mathbf{C} \in \mathcal{H}^k} |\mathcal{W}(\mathbf{C}, \mathbb{P}_n) - \mathcal{W}(\mathbf{C}, \mathbb{P})| \\
\leq & \frac{2}{n} \mathcal{R}(\mathcal{G}_{\mathbf{C}}) && \text{(by Eq.(16))} \\
= & \tilde{\mathcal{O}}\left(\sqrt{\frac{k}{n}}\right) && \text{(by Lemma 6).}
\end{aligned}
$$

This proves the result. $\qquad\square$

**Lemma 8.** *If constructing $\mathcal{I}$ by uniformly sampling*

$$m \geq C\sqrt{n}\log(1/\delta)\min(k, \Xi)/\sqrt{k},$$

*then for all $\mathcal{S} \in \mathcal{X}^n$, with probability at least $1 - \delta$, we have*

$$\mathcal{W}(\mathbf{C}_{n,m}, \mathbb{P}_n) - \mathcal{W}(\mathbf{C}_n, \mathbb{P}_n) \leq C\sqrt{\frac{k}{n}},$$

*where $\Xi = \mathrm{Tr}(\mathbf{K}_n(\mathbf{K}_n + \mathbf{I}_n)^{-1})$ is the effective dimension of $\mathbf{K}_n$, and $C$ is a constant.*

*Proof.* This can be directly proved by combining Lemma 1 and Lemma 2 of [2] by setting $\varepsilon = 1/2$. $\qquad\square$

*Proof of Theorem 4.* Note that

$$
\begin{aligned}
& \mathbb{E}[\mathcal{W}(\mathbf{C}_{n,m}, \mathbb{P})] - \mathcal{W}^*(\mathbb{P}) \\
= & \underbrace{\mathbb{E}[\mathcal{W}(\mathbf{C}_{n,m}, \mathbb{P}) - \mathcal{W}(\mathbf{C}_{n,m}, \mathbb{P}_n)]}_{A_1} + \underbrace{\mathbb{E}[\mathcal{W}(\mathbf{C}_{n,m}, \mathbb{P}_n) - \mathcal{W}(\mathbf{C}_n, \mathbb{P}_n)]}_{A_2} \\
& + \underbrace{\mathbb{E}[\mathcal{W}(\mathbf{C}_n, \mathbb{P}_n) - \mathcal{W}(\mathbf{C}_n, \mathbb{P})]}_{A_3} + \underbrace{\mathbb{E}[\mathcal{W}(\mathbf{C}_n, \mathbb{P})] - \mathcal{W}^*(\mathbb{P})}_{A_4}.
\end{aligned}
$$

Note that

$$
\begin{aligned}
A_3 \leq & \mathbb{E} \sup_{\mathbf{C} \in \mathcal{H}^k} \left|\mathcal{W}(\mathbf{C}, \mathbb{P}_n) - \mathcal{W}(\mathbf{C}, \mathbb{P})\right| \\
\leq & \frac{2}{n} \mathcal{R}(\mathcal{G}_{\mathbf{C}}) && \text{(by Eq.(16))} \\
\leq & \tilde{\mathcal{O}}\left(\sqrt{\frac{k}{n}}\right). && \text{(by Lemma 6)}
\end{aligned}
\qquad (20)
$$

One can see that $A_4$ can be bounded by $\tilde{\mathcal{O}}(\sqrt{k/n})$ using Theorem 1. $A_1$ and $A_2$ can both be bounded as $\tilde{\mathcal{O}}(\sqrt{k/n})$ using Lemma 7 and Lemma 8, respectively. $\qquad\square$

# Appendix: Proof of Theorem 5

*Proof.* From the definition of effective dimension, we have

$$
\begin{aligned}
\Xi =& \mathrm{Tr}(\mathbf{K}^{\mathrm{T}}(\mathbf{K}+\mathbf{I})^{-1}) = \sum_{i=1}^{n} \frac{\lambda_i}{\lambda_i + 1} \\
=& \sum_{i=1}^{\lfloor \sqrt{k} \rfloor} \frac{\lambda_i}{\lambda_i + 1} + \sum_{i=\lfloor \sqrt{k} \rfloor + 1}^{n} \frac{\lambda_i}{\lambda_i + 1} \leq \sum_{i=1}^{\lfloor \sqrt{k} \rfloor} 1 + \sum_{i=\lfloor \sqrt{k} \rfloor + 1}^{n} \lambda_i \\
\leq& \sqrt{k} + \sum_{i=\lfloor \sqrt{k} \rfloor + 1}^{n} \lambda_i \leq \sqrt{k} + \sum_{i=\lfloor \sqrt{k} \rfloor + 1}^{n} c i^{-\alpha} \\
\leq& \sqrt{k} + c \int_{\sqrt{k}}^{\infty} x^{-\alpha} dx = \sqrt{k} + \frac{c}{\alpha - 1} \sqrt{k}^{1-\alpha} \\
\leq& \left(1 + \frac{c}{\alpha - 1}\right) \sqrt{k}.
\end{aligned}
$$

Thus, we can obtain that

$$
\frac{\min(k, \Xi)}{\sqrt{k}} \leq \frac{\Xi}{\sqrt{k}} \leq 1 + \frac{c}{\alpha - 1}.
$$

Substituting the above inequality into Theorem 4, we can prove this result. $\qquad\square$

# Appendix: Proof of Theorem 6

*Proof.* Note that

$$
\begin{aligned}
& \mathbb{E}[\mathcal{W}(\tilde{\mathbf{C}}_{m,n}, \mathbb{P})] - \mathcal{W}^*(\mathbb{P}) \\
=& \underbrace{\mathbb{E}\left[\mathcal{W}(\tilde{\mathbf{C}}_{m,n}, \mathbb{P}) - \mathcal{W}(\tilde{\mathbf{C}}_{m,n}, \mathbb{P}_n)\right]}_{A_1} + \underbrace{\mathbb{E}\left[\mathcal{W}(\tilde{\mathbf{C}}_{m,n}, \mathbb{P}_n) - \mathcal{W}(\mathbf{C}_{m,n}, \mathbb{P}_n)\right]}_{A_2} \\
& + \underbrace{\mathbb{E}\left[\mathcal{W}(\mathbf{C}_{m,n}, \mathbb{P}_n) - \mathcal{W}(\mathbf{C}_{m,n}, \mathbb{P})\right]}_{A_3} + \underbrace{\mathbb{E}\left[\mathcal{W}(\mathbf{C}_{m,n}, \mathbb{P})\right] - \mathcal{W}^*(\mathbb{P})}_{A_4}.
\end{aligned}
$$

Also note that $A_2$ is bounded by $\zeta$, $A_4$ can be obtained from Theorem 5, and $A_1$ and $A_3$ can be bounded by the Rademacher complexity:

$$
A_1, A_3 \leq \mathbb{E} \sup_{\mathbf{C} \in \mathcal{H}^k} |\mathcal{W}(\mathbf{C}, \mathbb{P}_n) - \mathcal{W}(\mathbf{C}, \mathbb{P})| \leq \frac{2}{n} \mathcal{R}(\mathcal{G}_{\mathbf{C}}).
$$

Thus, we can obtain that

$$
\mathbb{E}[\mathcal{W}(\tilde{\mathbf{C}}_n, \mathbb{P})] - \mathcal{W}^*(\mathbb{P}) = \tilde{\mathcal{O}}\left(\frac{\mathcal{R}(\mathcal{G}_{\mathbf{C}})}{n} + \sqrt{\frac{k}{n}} + \zeta\right). \tag{21}
$$

Substituting Lemma 6 into Eq. (21), we can proves the result. $\qquad\square$

# Appendix: Proof of Theorem 7

*Proof.* Note that

$$
\mathbb{E}\left[\mathbb{E}_{\mathcal{A}}[\mathcal{W}(\mathbf{C}_{n,m}^{\mathcal{A}}, \mathbb{P})]\right] = \mathbb{E}\left[\mathbb{E}_{\mathcal{A}}[\mathcal{W}(\mathbf{C}_{n,m}^{\mathcal{A}}, \mathbb{P})] - \mathbb{E}_{\mathcal{A}}[\mathcal{W}(\mathbf{C}_{n,m}^{\mathcal{A}}, \mathbb{P}_n)]\right] + \mathbb{E}\left[\mathbb{E}_{\mathcal{A}}[\mathcal{W}(\mathbf{C}_{n,m}^{\mathcal{A}}, \mathbb{P}_n)]\right].
$$

By Lemma 2, we can obtain that

$$
\begin{aligned}
& \mathbb{E}\left[\mathbb{E}_{\mathcal{A}}[\mathcal{W}(\mathbf{C}_{n,m}^{\mathcal{A}}, \mathbb{P}_n)]\right] \leq \beta \cdot \mathbb{E}\left[\mathcal{W}(\mathbf{C}_{n,m}, \mathbb{P}_n)\right] \\
=& \beta \cdot \mathbb{E}\left[\mathcal{W}(\mathbf{C}_{n,m}, \mathbb{P}_n) - \mathcal{W}(\mathbf{C}_{n,m}, \mathbb{P})\right] + \beta \cdot \mathbb{E}\left[\mathcal{W}(\mathbf{C}_{n,m}, \mathbb{P})\right].
\end{aligned}
$$

Thus, we can obtain that

$$
\mathbb{E}\Big[\mathbb{E}_{\mathcal{A}}[\mathcal{W}(\mathbf{C}_{n,m}^{\mathcal{A}}, \mathbb{P})]\Big]
$$
$$
\leq \underbrace{\mathbb{E}\Big[\mathbb{E}_{\mathcal{A}}[\mathcal{W}(\mathbf{C}_{n,m}^{\mathcal{A}}, \mathbb{P})] - \mathbb{E}_{\mathcal{A}}[\mathcal{W}(\mathbf{C}_{n,m}^{\mathcal{A}}, \mathbb{P}_{n,m})]\Big]}_{A_1}
$$
$$
+ \beta \cdot \underbrace{\mathbb{E}\Big[\mathcal{W}(\mathbf{C}_{n,m}, \mathbb{P}_{n,m}) - \mathcal{W}(\mathbf{C}_{n,m}, \mathbb{P})\Big]}_{A_2} + \beta \cdot \underbrace{\mathbb{E}\Big[\mathcal{W}(\mathbf{C}_{n,m}, \mathbb{P})\Big]}_{A_3}.
$$

Note that

$$
A_1, A_2 \leq \mathbb{E}\sup_{\mathbf{C}\in\mathcal{H}^k} \big|\mathcal{W}(\mathbf{C}, \mathbb{P}_n) - \mathcal{W}(\mathbf{C}, \mathbb{P})\big|
$$
$$
\leq \frac{2}{n}\mathcal{R}(\mathcal{G}_{\mathbf{C}}) \qquad\qquad \text{(by Eq. (16))}
$$
$$
= \tilde{\mathcal{O}}\left(\sqrt{\frac{k}{n}}\right) \qquad\qquad \text{(by Lemma 6).}
$$

By Corollary 5, $A_3$ can be bounded:

$$
A_3 = \mathbb{E}[\mathcal{W}(\mathbf{C}_{n,m}, \mathbb{P})] \leq \mathcal{W}^*(\mathbb{P}) + c\sqrt{\frac{k}{n}}\log^2\left(\sqrt{n}\right).
$$

This proves the result. $\qquad\qquad\qquad\qquad\qquad\qquad\qquad\qquad\qquad\qquad\qquad\square$

## Appendix: Proof of Lemma 2

We first prove that the maximum Rademacher complexity can be bounded by $3\sqrt{n}$. Then, following the same idea as [5] and using the Khintchine inequality [6], we show that there exists a hypothesis function $\mathcal{F}_{\mathbf{C}}$ such that $\mathcal{R}_n(\mathcal{G}_{\mathbf{C}}) \geq \sqrt{\frac{kn}{2}}$.

**Lemma 9** (Khintchine inequality with $p = 1$ in [6]). *Let $\sigma_1, \ldots, \sigma_n$ be Rademacher variables with equal probability of taking values $+1$ or $-1$. Then, we have $\mathbb{E}_{\boldsymbol{\sigma}}|\sum_{i=1}^n \sigma_i| \geq \sqrt{\frac{n}{2}}$.*

*Proof of Lemma 2.* Let $\epsilon_1, \ldots, \epsilon_k$ be independent random variables with equal probability of taking values $+1$ or $-1$. Let $\mathbf{C} = (\epsilon_1\boldsymbol{\nu}_1, \ldots, \epsilon_k\boldsymbol{\nu}_k)$, where $\boldsymbol{\nu}_i$ is the $i$th standard basis function in $\mathcal{H}$, that is $\langle\boldsymbol{\nu}_i, \boldsymbol{\nu}_j\rangle = 1$ if $i = j$, otherwise 0. We choose the hypothesis space

$$
\mathcal{F}_{\mathbf{C}} = \left\{ f_{\mathbf{C}} = (f_{\epsilon_1\boldsymbol{\nu}_1}, \ldots, f_{\epsilon_k\boldsymbol{\nu}_k}) \Big| f_{\epsilon_i\boldsymbol{\nu}_i}(\mathbf{x}) = \|\Phi_{\mathbf{x}} - \epsilon_i\boldsymbol{\nu}_i\|^2, \boldsymbol{\epsilon}\in\{\pm 1\}^k \right\}. \tag{22}
$$

Assume that $n$ is divisible by $k$. We set $\Phi_1, \ldots, \Phi_{n/k} = \boldsymbol{\nu}_1, \Phi_{(n+1)/k}, \ldots, \Phi_{2n/k} = \boldsymbol{\nu}_2, \ldots$, and so on, and let $i_t$ be the index such that $\Phi_t = \boldsymbol{\nu}_{i_t}$. Let $\boldsymbol{\sigma}' \in \{\pm 1\}^n$ be Rademacher variables. From the

definition of clustering Rademacher complexity, we can obtain that

$$\mathcal{R}_n(\mathcal{G}_{\mathbf{C}}) = \mathcal{R}_n(\varphi \circ \mathcal{F}_{\mathbf{C}})$$

$$= \mathbb{E}_{\boldsymbol{\sigma}' \in \{\pm 1\}^n} \sup_{\boldsymbol{\epsilon} \in \{\pm 1\}^k} \left| \sum_{t=1}^n \sigma'_t \min_{1 \le i \le k} \|\Phi_t - \epsilon_i \boldsymbol{\nu}_i\|^2 \right|$$

$$= \mathbb{E}_{\boldsymbol{\sigma}' \in \{\pm 1\}^n} \sup_{\boldsymbol{\epsilon} \in \{\pm 1\}^k} \left| \sum_{t=1}^n \sigma'_t \min_{1 \le i \le k} (2 - 2\langle \Phi_t, \epsilon_i \boldsymbol{\nu}_i \rangle) \right|$$

(since $\Phi_t = \boldsymbol{\nu}_{i_t}$ and $\boldsymbol{\nu}_i$ is the $i$th standard basis function in $\mathcal{H}$)

$$= 2 \mathbb{E}_{\boldsymbol{\sigma}' \in \{\pm 1\}^n} \sup_{\boldsymbol{\epsilon} \in \{\pm 1\}^k} \left| \sum_{t=1}^n \sigma'_t \max_{1 \le i \le k} \langle \Phi_t, \epsilon_i \boldsymbol{\nu}_i \rangle \right|$$

$$= 2 \mathbb{E}_{\boldsymbol{\sigma}' \in \{\pm 1\}^n} \sup_{\boldsymbol{\epsilon} \in \{\pm 1\}^k} \left| \sum_{t=1}^n \sigma'_t \max\{\epsilon_{i_t}, 0\} \right| \tag{23}$$

$$\ge 2 \mathbb{E}_{\boldsymbol{\sigma}' \in \{\pm 1\}^n} \sup_{\boldsymbol{\epsilon} \in \{\pm 1\}^k} \sum_{t=1}^n \sigma'_t \max\{\epsilon_{i_t}, 0\}$$

$$= 2k \cdot \mathbb{E}_{\boldsymbol{\sigma}' \in \{\pm 1\}^{n/k}} \sup_{\epsilon \in \{\pm 1\}} \sum_{t=1}^{n/k} \sigma'_t \max\{\epsilon, 0\}$$

$$= 2k \cdot \frac{1}{2} \mathbb{E}_{\boldsymbol{\sigma}' \in \{\pm 1\}^{n/k}} \left| \sum_{t=1}^{n/k} \sigma'_t \right| \ge k \sqrt{\frac{n}{2k}} \quad \text{(by Lemma 9)}$$

$$= \sqrt{\frac{nk}{2}}.$$

From Lemma 5, we know that

$$\max_i \tilde{\mathcal{R}}_n(\mathcal{F}_{\mathbf{C}_i}) \le 3\sqrt{n}.$$

Thus, by the above upper bounds the lower bound (Eq.(23)), we can prove that there exists a hypothesis space $\mathcal{F}_{\mathbf{C}}$ defined in (22), such that

$$\mathcal{R}_n(\mathcal{G}_{\mathbf{C}}) \ge \frac{\sqrt{k}}{3\sqrt{2}} \cdot \max_i \tilde{\mathcal{R}}_n(\mathcal{F}_{\mathbf{C}_i}).$$

This proves the result. □