# OpenReview forum: "Refined Learning Bounds for Kernel and Approximate $k$-Means"
_NeurIPS.cc/2021/Conference — NeurIPS 2021 Spotlight_

### Official Review · Reviewer_Cavx · 2021-07-11

**Rating:** 8
**Confidence:** 4

**Summary:**

The paper provides a nearly optimal bound for consistency of kernel $k$-means clustering and gives an approximation scheme based on the Nystrom method that allows one to perform scalable data quantization. More specifically, a new upper bound in $\sqrt{\frac{k}{n}}$ is obtained for the excess risk in kernel $k$-means, which improves the state-of-the-art excess risk bound in $\frac{k}{\sqrt{n}}$. The same bound for the Nystrom approximation of the algorithm are also provided.

**Ethical Concerns:**

None.

**Limitations And Societal Impact:**

Here are some comments:
1. It would be better to provide more intuition and clear descriptions of the proofs and the relations among them can help the reader better understand the content.
2. In Line 207, the paper mentions "improved $k$-kernel means++". What's this? Is this improved $k$-means++ ? The paper should provide a brief introduction about this.
3. It would be better to introduce the experimental results in more detail.

**Main Review:**

The paper proposes a nearly optimal excess clustering risk bound is proposed for ERM, which is the first (nearly) optimal excess risk bound for kernel $k$-means in terms of both $k$ and $n$. It improves the bound from $\frac{k}{\sqrt{n}}$ to $\sqrt{\frac{k}{n}}$, which is a significant advancement. Moreover, the general cases beyond ERM are also considered. The paper also considers the computational aspects, i.e., the paper states the same bound for Nystrom approximation of the algorithm. These results are important for kernel $k$-means and are very interesting. The paper also provides some numerical experiments to demonstrate the theoretical results. In summary, I think this is a good and well-written theoretical paper which is helpful to who are interested in kernel methods or kernel $k$-means.

**Time Spent Reviewing:**

4 days.

---

> ### Author Response · Authors · 2021-08-08
> **Response to Reviewer Cavx**
>
> Thanks for your recognition of this paper and the useful comments and suggestions,  we will take them all into account to improve the presentation.
>
> 1. We will provide more intuition and clear descriptions of the proofs and the relations among them the final version. Thanks for your suggestion.
> 2. The 'improved kernel means++' is a new iterative algorithms to obtain the approximate solution of $k$-means++ proposed in [25], it uses a simple combination of k-means++ sampling and a local search strategy,  which has been shown that the empirical squared norm criterion of the approximate solution  can be up to a constant factor from the optimal empirical solution. We will give an overview of the improved kernel means++ in supplementary in the final version. Thanks for your suggestion.
> 3. We will introduce the experimental results in more detail in the final version. Thanks for your suggestion.

---

### Official Review · Reviewer_vjeN · 2021-07-14

**Rating:** 6
**Confidence:** 1

**Summary:**

The authors proposed optimal risk bound for clustering using kernel kmeans and Nystrom based kernel k-means. They improve the risk bound from k/sqrt(n) to sqrt(k/n).

**Limitations And Societal Impact:**

1. An overview of the algorithm in the main paper or supplementary would be beneficial.
2. Some more experiments on larger real data set would have been better.
3. Regrading the sample size in theorem-4 & 5: The notion of statistical dimension is also used to show importance sampling-based spectral property approximation "Avron H, Clarkson KL, Woodruff DP. Sharper Bounds for Regularized Data Fitting. Approximation, Randomization, and Combinatorial Optimization. Algorithms and Techniques. 2017 Aug". Do you think a similar sampling technique can be used here to get a better sample size?
4. Although the authors acknowledge about "discussing limitations" however, I did not find any discussion.

**Main Review:**

1. The paper is well organized and well written.
2. The theoretical results are significant as the author shows near-optimal risk bounds.


----
I appreciate for the response and increase my final score.

**Time Spent Reviewing:**

5-6

---

> ### Author Response · Authors · 2021-08-08
> **Response to Reviewer vjeN**
>
> Thanks for your useful comments and suggestions, we will take them all into account to improve the presentation.
>
> 1.  Becasue of the focus of this manuscript is on the theoretical analysis of kernel $k$-means and approximated $k$-means,  and these algorithms does indeed exist in [9, 25],  we omit the description of these algorithms and only illustrate that the  reader can refer to [9, 25] for more detail in the previous version.  In order to increase readability, we promise that we will add an overview of the algorithm in supplementary in the final version. Thanks for your suggestion.
> 2. To  reflect real-world scenarios on large real data, we add more experiments on the real data sets. The empirical evaluations with Gaussian kernel $\exp(- \|x-x'\|^2/\sigma^2)$,  $\sigma=\sqrt{\sum_{ij}\|x_i-x_j\|^2}/n$, are given in the following table:
>
> | Dataset | Datasize| Kernel $k$-means| Nystr\"{o}m Kernel $k$-Means ($m=100$) |
> |:-------:|:--------:|:--------:|:----------:|
> |dna  |  2000 | 0.53  | 0.52 |
> |segment|  2310| 0.55 |  0.55|
> |mushrooms| 8124| 0.66| 0.65|
> |mnist| 60000 | --    | 0.43 |
> |skin-nonskin|  245057| --|0.63|
> |covtype |581012| --  | 0.32 |
>
> From the above results on real data sets, one can see that, on the large data sets, such as mnist, skin-nonskin and covtype,  Nystr\"{o}m kernel $k$-means can be also  used to address, which indicates the effective of this approximate kernel $k$-means.
>
> 3. Compared with the Lemma 1 of  [Calandriello 2018] and  Theorem 3 of [Avron 2017], if $k<\epsilon^{-1}\Xi$, the dominant terms of [Calandriello 2018] and [Avron 2017] are both $\Omega\left(\frac{\Xi}{\epsilon^2}\right)$,  thus we think a similar sampling technique of [Avron 2017] used here may also get a similar (but not better) sample size. Thank you for raising such a good research question, we will explore this problem in our future work.
>
> [Avron 2017] Avron H, Clarkson KL, Woodruff DP. Sharper Bounds for Regularized Data Fitting. Approximation, Randomization, and Combinatorial Optimization. Algorithms and Techniques. 2017
>
> [Calandriello 2018] Calandriello D and Rosasco L. Statistical and Computational Trade-Offs in Kernel K-Means. In NeurIPS, 2018.
>
> 4. The main limitation we think is that we only derived the risk bounds of learning rate $\mathcal{O}(\sqrt{k/n})$ for the basic case.  In the future, we will consider studying whether it is possible to prove a bound of $\mathcal{O}(\sqrt{k}/n)$ under **certain strict assumptions**. We will add this discussion of the limitation in the final version. Thanks for your suggestion.

---

### Official Review · Reviewer_SaBc · 2021-07-15

**Rating:** 7
**Confidence:** 3

**Summary:**

In this paper, the authors provide a near-optimal risk bound for kernel $k$-means, namely $\tilde{O}(\sqrt{k/n})$, which improves upon the existing risk bound $O(k/\sqrt{n})$, and nearly matches the lower bound $\Omega(\sqrt{k/n})$, where $k$ is the number of clusters and $n$ is the size of the training set. They first derive a near-optimal risk bound for the empirical risk minimizer (ERM) and then extend it to general cases beyond ERM and $k$-means++. For efficiency, the authors also provide near-optimal risk bounds for Nyström kernel $k$-means for the case of using $\tilde{\Omega}(\sqrt{n})$ sampling points. The authors have performed numerical experiments to validate their theoretical findings.

**Limitations And Societal Impact:**

As mentioned in the Checklist, the limitations have been addressed and potential negative societal impact is not applicable.

**Main Review:**

My impression is that this is an interesting work with solid theoretical contributions, so overall I am inclined to accept. Some minor comments are listed as follows:

1. Mentioning that $k$ is the number of clusters and $n$ is the size of the training set in the abstract.

2. The authors should check all the notations used in the main document carefully. For example, in Section 2.1, the definition of the empirical distribution $\mathbb{P}_n(\mathcal{S})$ at Line 73 is problematic because it always equals $1$. Line 82, $\mu = \Omega(\nu)$ does not mean $c_1 \nu \le \mu \le c_2 \nu$.

3. For brevity, some notations/equations that are never used in the main body can be omitted or be placed in the supplementary material. For example, \mathcal{W}(\mathbf{C}, \mathbb{P}_n)  does not need to be expanded as the equation between Eqs. (1) and (2). The definitions of f_{\mathbf{C}} and \mathcal{F}_{\mathbf{C}} (Eq. (3)) are never used in the main document. In fact, I think the content between Lines 92 and 98 can all be removed (or move to the supplementary material). The notation $\mathcal{E}(\mathbf{C}_n)$ (excess clustering risk) has never been used in the statements of the theorems.

4. Line 103, the authors should not use $\mathbf{C}$ to represent the collection of centroids corresponding to the lower bound. Perhaps renaming it as $\mathbf{C}_{lb}$. The authors should add "for example" before presenting Lemma 1.

5. Between Lines 170 and 171, removing  "span" in the definition of $\mathcal{H}_m$.

6. Checking the format of all the references in the References section. For example, in [2], it should be "NP-hardness of Euclidean". In [8], "risk" should be "Risk".

**Time Spent Reviewing:**

8

---

> ### Author Response · Authors · 2021-08-08
> **Response to Reviewer SaBc**
>
> Thanks for your recognition of this paper and the useful comments and suggestions, we will take them all into account to improve the presentation. We will carefully proofread the paper to eliminate all typos in the final vesion.
>
> 1. We will mention that $k$ is the number of clusters and $n$ is the size of the training set in the abstract in the final version. Thanks for your suggestion.
> 2. The definition of the empirical distribution should be $\mathbb{P}_n(\mathbf x)=\frac{1}{n}$ if $\mathbf x\in\mathcal{S}$, otherwise 0. We will use $\mu\simeq \nu$ to mean $c_1\nu\leq \mu\leq c_2\nu$. We will correct these in the final version.
> 3. We wll check the full manuscript to remove the notations/equations that are never used in the main body to the supplementary material for brevity in the final version. Thanks for your suggestion.
> 4. We will resplace $\mathbf C$ as $\mathbf C_{lb}$ to represent the collection of centroids corresponding to the lower bound. We will add  ``for example" before presenting Lemma 1.
> 5. We will  remove the 'span' in the defintion of $\mathcal{H}_m$  between lines 170 and 171.
> 6. We will check the format of all the references in the References section in the final version.

---

> > ### Comment · Reviewer_SaBc · 2021-08-24
> > **Thanks for the authors responses**
> >
> > After reading the responses and other reviewers' comments, I still think that this is an interesting work with solid theoretical contributions. I would like to keep my original score (though I have a little ethical concern about the arxiv version of this submission since the titles are different).

---

> > > ### Author Response · Authors · 2021-08-24
> > > **Thank you for your appreciation of this manuscript.**
> > >
> > > Thanks for your recognition of this manuscript. As there are some additional results added and also for better anonymous review, a different title of this manuscript is adopted. According to the policy of NeurIPS 2021 on Preprints, that is "Authors may submit anonymized work to NeurIPS that is already available as a preprint (e.g., on arXiv) without citing it", so we think it's allowed to use different title and can make the title consistent if it's necessary.

---

### Official Review · Reviewer_uDpd · 2021-07-16

**Rating:** 7
**Confidence:** 4

**Summary:**

This manuscript derived nearly optimal risk bounds for both kernel k-means and Nystrom kernel k-means of learning rate of O(sqrt(k/n)), which is interesting and attractive.

**Limitations And Societal Impact:**

Cons:
1. Weak innovation, all contributions and most of the results of this manuscript can also be found in the journal: Nearly Optimal Clustering Risk Bounds for Kernel K-Means, Machine Learning, arXiv:2003.03888.
2. The Lemma3 mentioned in the main result and the Lemma10 mentioned in section 4.2 are not found in the script, making the result of this article unreliable.
3. Weak experiments. The employed datasets are synthetic , clearly unable to reflect real-world scenarios.

**Main Review:**

Pros:
1. Well-organized and clearly written, full and accurate understanding of related works.
2. A (nearly) optimal excess clustering risk bound is proposed in this paper, which is the main contribution of it.

**Time Spent Reviewing:**

tw0

---

> ### Author Response · Authors · 2021-08-08
> **Response to Reviewer uDpd**
>
> 1. From the policy of NeurIPS 2021 on Preprints, that is "the existence of non-anonymous preprints (on **arXiv** or other online repositories, personal websites, social media)  **will not result in rejection**", thus, all contributions and most of the results can be found in the arXiv cannot be the reason for the rejection of our manuscript.
> 2. Lemma 3 and Lemma 10 can be found in the **supplementary material**. In order to increase the readability of this manuscript, we will add the annotates of the results that given in supplementary material in the final version.
> 3. This manuscript focuses on theoretical analysis of kernel $k$-means and approximated $k$-means,  thus we only consider the use of the synthetic experiments to verify the theoretical findings.
> To  reflect real-world scenarios, we add more experiments on the real data sets. We use 6 publicly avaiable datasets, dna, segment, mushrooms, mnist, skin-nonskin and covtype, from the LIBSVM Data. The empirical evaluations with Gaussian kernel $\exp(- \|x-x'\|^2/\sigma^2)$,  $\sigma=\sqrt{\sum_{ij}\|x_i-x_j\|^2}/n$, are given in the following table:
> | Dataset | Datasize| Kernel $k$-means| Nystr\"{o}m Kernel $k$-Means ($m=100$) |
> |:-------:|:--------:|:--------:|:----------:|
> |dna  |  2000 | 0.53  | 0.52 |
> |segment|  2310| 0.55 |  0.55|
> |mushrooms| 8124| 0.66| 0.65|
> |mnist| 60000 | --    | 0.43 |
> |skin-nonskin|  245057| --|0.63|
> |covtype |581012| --  | 0.32 |
>
> From the above results on real data sets, we can find that Nystr\"{o}m kernel $k$-means give the similar results as the original one, which also match the theoretical findings.

---

### Author Response · Authors · 2021-08-08
**Thanks for the four anonymous reviewers**

We would like to thank four anonymous reviewers for their valuable comments and rapid responses, and their useful comments and suggestions help us improve the manuscript greatly. In the following, we attempt to answer all the questions involved in the review report. The following  is a point-to-point response to the four reviewers.

---

### Decision · Program_Chairs · 2021-09-27

**Decision:**

Accept (Spotlight)

**Comment:**

The work presents nearly optimal bounds for the excess clustering risk of kernel k-means, improving the rate of convergence from k/sqrt(n) to sqrt(k/n), and also shows that approximate (Nystrom with uniform sampling of sqrt(nk) landmark points) has the same order of excess clustering risk. This theoretical result is a strong contribution and of great relevance to the community.